# Arginine-vasopressin mediates counter-regulatory glucagon release and is diminished in type 1 diabetes

Angela Kim[1,2], Jakob G Knudsen[3,4], Joseph C Madara[1], Anna Benrick[5], Thomas G Hill[3], Lina Abdul Kadir[3], Joely A Kellard[3], Lisa Mellander[5], Caroline Miranda[5], Haopeng Lin[6], Timothy James[7], Kinga Suba[8], Aliya F Spigelman[6], Yanling Wu[5], Patrick E MacDonald[6], Ingrid Wernstedt Asterholm[5], Tore Magnussen[9], Mikkel Christensen[9,10,11], Tina Vilsbøll[9,10,12], Victoria Salem[8], Filip K Knop[9,10,12,13], Patrik Rorsman[3,5], Bradford B Lowell[1,2], Linford JB Briant[3,14]*

[1]Division of Endocrinology, Diabetes, and Metabolism, Beth Israel Deaconess Medical Center, Boston, United States; [2]Program in Neuroscience, Harvard Medical School, Boston, United States; [3]Oxford Centre for Diabetes, Endocrinology and Metabolism, Radcliffe Department of Medicine, University of Oxford, Oxford, United Kingdom; [4]Section for Cell Biology and Physiology, Department of Biology, University of Copenhagen, Copenhagen, Denmark; [5]Metabolic Research Unit, Institute of Neuroscience and Physiology, Sahlgrenska Academy at University of Gothenburg, Göteborg, Sweden; [6]Alberta Diabetes Institute, Li Ka Shing Centre for Health Research Innovation, Edmonton, Canada; [7]Department of Clinical Biochemistry, John Radcliffe, Oxford NHS Trust, Oxford, United Kingdom; [8]Section of Cell Biology and Functional Genomics, Department of Metabolism, Digestion and Reproduction, Imperial College London, London, United Kingdom; [9]Center for Clinical Metabolic Research, Gentofte Hospital, Hellerup, Denmark; [10]Department of Clinical Pharmacology, Bispebjerg Hospital, University of Copenhagen, Copenhagen, Denmark; [11]Department of Clinical Medicine, Faculty of Health and Medical Sciences, University of Copenhagen, Copenhagen, Denmark; [12]Steno Diabetes Center Copenhagen, Copenhagen, Denmark; [13]Novo Nordisk Foundation Center for Basic Metabolic Research, Faculty of Health and Medical Sciences, University of Copenhagen, Copenhagen, Denmark; [14]Department of Computer Science, University of Oxford, Oxford, United Kingdom

*For correspondence:
linford.briant@ocdem.ox.ac.uk

**Competing interest:** The authors declare that no competing interests exist.

**Abstract** Insulin-induced hypoglycemia is a major treatment barrier in type-1 diabetes (T1D). Accordingly, it is important that we understand the mechanisms regulating the circulating levels of glucagon. Varying glucose over the range of concentrations that occur physiologically between the fed and fuel-deprived states (8 to 4 mM) has no significant effect on glucagon secretion in the perfused mouse pancreas or in isolated mouse islets (in vitro), and yet associates with dramatic increases in plasma glucagon. The identity of the systemic factor(s) that elevates circulating glucagon remains unknown. Here, we show that arginine-vasopressin (AVP), secreted from the posterior pituitary, stimulates glucagon secretion. Alpha-cells express high levels of the vasopressin 1b receptor (V1bR) gene (*Avpr1b*). Activation of AVP neurons in vivo increased circulating copeptin (the C-terminal segment of the AVP precursor peptide) and increased blood glucose; effects blocked by pharmacological antagonism of either the glucagon receptor or V1bR. AVP also mediates the stimulatory

effects of hypoglycemia produced by exogenous insulin and 2-deoxy-D-glucose on glucagon secretion. We show that the A1/C1 neurons of the medulla oblongata drive AVP neuron activation in response to insulin-induced hypoglycemia. AVP injection increased cytoplasmic $Ca^{2+}$ in alpha-cells (implanted into the anterior chamber of the eye) and glucagon release. Hypoglycemia also increases circulating levels of AVP/copeptin in humans and this hormone stimulates glucagon secretion from human islets. In patients with T1D, hypoglycemia failed to increase both copeptin and glucagon. These findings suggest that AVP is a physiological systemic regulator of glucagon secretion and that this mechanism becomes impaired in T1D.

## Editor's evaluation

The authors have revised their manuscript in response to the comments and suggestions by the two reviewers. The current study provides compelling data to support a mechanistic model that the CNS regulates glucagon secretion through glucose-regulated AVP release.

## Introduction

Glucagon is secreted from alpha-cells of the pancreatic islets and has many physiological actions; most notably the potent stimulation of hepatic glucose production to restore euglycemia when blood glucose has fallen below the normal range (a process referred to as counter-regulation). The importance of glucagon for glucose homeostasis is well established (**Frayn and Evans, 2019**). In both type-1 diabetes (T1D) and type-2 diabetes (T2D), hyperglycemia results from a combination of complete/partial loss of insulin secretion and over-secretion of glucagon (**Unger and Cherrington, 2012**). In addition, counter-regulatory glucagon secretion becomes impaired in both forms of diabetes (particularly T1D), which may result in life-threatening hypoglycemia (**UK Hypoglycaemia Study Group, 2007**). Despite the centrality of glucagon to diabetes etiology, there remains considerable uncertainty about the regulation of its release and the relative importance of intra-islet effects and systemic factors (**Gylfe, 2013**; **Lai et al., 2018**). Based on observations in isolated (ex vivo) islets, hypoglycemia has been postulated to stimulate glucagon secretion via intrinsic (**Yu et al., 2019**; **Rorsman et al., 2014**; **Basco et al., 2018**) and/or paracrine mechanisms (**Briant et al., 2018b**; **Vergari et al., 2019**). While it is indisputable that the islet is a critical component of the body's 'glucostat' (**Rodriguez-Diaz et al., 2018**) and has the ability to intrinsically modulate glucagon output, it is clear that such an 'islet-centric' viewpoint is overly simplistic (**Schwartz et al., 2013**). Indeed, many studies have clearly demonstrated that brain-directed interventions can profoundly alter islet alpha-cell function, with glucose-sensing neurons in the hypothalamus being key mediators (**Garfield et al., 2014**; **Lamy et al., 2014**; **Meek et al., 2016**; **Flak et al., 2020**). This ability of the brain to modulate glucagon secretion is commonly attributed to the autonomic innervation of the pancreas (**Lamy et al., 2014**; **Marty et al., 2005**; **Thorens, 2014**). However, glucagon secretion is not only restored in pancreas transplantation patients but also insensitive to adrenergic blockade (**Barrou et al., 1994**; **Diem et al., 1990**), suggesting that other (non-autonomic) central mechanisms may also regulate glucagon secretion in vivo.

Arginine-vasopressin (AVP) is a hormone synthesized in the hypothalamus (reviewed in **Bourque, 2008**). AVP neurons are divided into two classes: parvocellular AVP neurons (which either project to the median eminence to stimulate ACTH and glucocorticoid release or project centrally to other brain regions) and magnocellular AVP neurons (which are the main contributors to circulating levels of AVP). The parvocellular neurons reside solely in the paraventricular nucleus of the hypothalamus (PVH), whereas magnocellular neurons are found in both the PVH and supraoptic nucleus (SON). Stimulation of the magnocellular AVP neurons causes the release of AVP into the posterior pituitary, where it enters the peripheral circulation.

Under normal conditions, *Avpr1b* is one of the most enriched transcripts in alpha-cells from both mice (**van der Meulen et al., 2017**; **Lawlor et al., 2017**) and humans (**Nica et al., 2013**; **Blodgett et al., 2015**). This raises the possibility that AVP may be an important regulator of glucagon secretion under physiological conditions. Indeed, the ability of exogenous AVP and AVP analogs to potently stimulate glucagon secretion ex vivo has been known for some time (**Dunning et al., 1984**). However, whether circulating AVP contributes to physiological counter-regulatory glucagon release and how this regulation is affected in diabetes remains unknown.

Here, we have investigated the regulation of glucagon by circulating AVP. We first explored the role of AVP during hypoglycemia, a potent stimulus of glucagon secretion. Next, we explored the link between AVP and glucagon in humans and provided evidence that this putative 'hypothalamic-alpha-cell axis' is impaired in T1D.

## Results

### AVP evokes hyperglycemia and hyperglucagonemia

We first investigated the metabolic effects of AVP in vivo (*Figure 1a–c*). To do this, we took a 'designer receptors exclusively activated by designer drugs' (DREADD) approach. This chemogenetic technology allows cell-type-specific manipulation of neuronal function (*Roth, 2016*). We expressed the modified human M3 muscarinic receptor hM3Dq (see *Alexander et al., 2009*) in AVP neurons by bilaterally injecting a Cre-dependent virus-containing hM3Dq (AAV-DIO-hM3Dq-mCherry) into the SON of mice bearing *Avp* promoter-driven Cre recombinase (*Avp*[ires-Cre+] mice; *Figure 1a*). Expression of hM3Dq was limited to the SON (*Figure 1—figure supplement 1a-c*), thus allowing targeted activation of magnocellular AVP neurons (that release AVP into the circulation). Patch-clamp recordings confirmed that bath application of clozapine-N-oxide (CNO; 5–10 μM)—a specific, pharmacologically inert agonist for hM3Dq—induced membrane depolarisation and increased the firing rate in hM3Dq-expressing AVP neurons (*Figure 1—figure supplement 1d, e*). Injection of CNO (3 mg/kg i.p.) in vivo increased blood glucose (*Figure 1b*). We measured copeptin, the C-terminal segment of the AVP precursor peptide, which is a stable surrogate marker for AVP (*Morgenthaler et al., 2008*; *Christ-Crain and Fenske, 2016*), but the sample volume requirements (100 μl plasma) only allowed a single (terminal) measurement. Despite these experimental limitations, copeptin was elevated in response to CNO compared to saline injection (*Figure 1c*). It is notable that copeptin is a much larger peptide than AVP and its clearance from the circulation is slower (*Fenske et al., 2018*). Thus, circulating AVP levels may undergo more dramatic variations (see *Roussel et al., 2014* and data shown below).

To establish the contribution of glucagon to this hyperglycemic response of SON AVP neuron activation, we pre-treated mice with the glucagon receptor antagonist LY2409021 (*Kazda et al., 2016*). This completely abolished the hyperglycemic action of CNO (*Figure 1b*). Similarly, to understand the contribution of vasopressin 1b receptor (V1bR) signaling, we pre-treated mice with the V1bR antagonist SSR149415 (*Serradeil-Le Gal et al., 2002*). This also abolished the hyperglycemic effect of CNO (*Figure 1b*), suggesting that V1bR signaling mediates this response. CNO did not change food intake (*Figure 1—figure supplement 1f*) and did not have an off-target effect on blood glucose in *Avp*[ires-Cre/+] mice expressing a passive protein (mCherry) under AAV transfection in the SON (*Figure 1—figure supplement 1g*). Exogenous AVP also caused an increase in glucose (measured by continuous glucose monitoring [CGM] or standard blood sampling) and glucagon relative to saline injection in wild-type mice (*Figure 1—figure supplement 1h-j*).

### Insulin-induced glucagon secretion in vivo is driven by AVP

We next investigated whether AVP stimulates glucagon secretion during hypoglycemia in vivo. Insulin-induced hypoglycemia (from 8 mM to 4 mM) increased circulating glucagon levels tenfold (*Figure 1d and e*). The same decrease in extracellular glucose only marginally stimulated glucagon secretion measured in the perfused mouse pancreas (*Figure 1f*). Similarly, there was no difference in glucagon secretion measured at 8 mM and 4 mM glucose during 60 min static incubations of isolated mouse islets (*Figure 1g*). This glucose dependence of glucagon secretion in isolated islets and the perfused pancreas is in keeping with numerous other reports (*Lai et al., 2018*; *Salehi et al., 2006*; *Capozzi et al., 2019*; *Gerich et al., 1974*; *Walker et al., 2011*; *Briant et al., 2018a*), including 'perifused islets' (*Capozzi et al., 2019*). Therefore, additional mechanisms extrinsic to the islet clearly participate in the control of glucagon secretion in vivo.

We hypothesised that this extrinsic stimulus involves AVP. We investigated (using in vivo fiber photometry) whether AVP neuron activity is increased in response to an ITT (*Figure 1h*). We simultaneously recorded AVP neuron activity and plasma glucose (by CGM; *Figure 1i and j*). This revealed that insulin evoked an initial peak in AVP neuron activity followed by sustained activity (*Figure 1i*). The initial peak occurred when glucose had fallen to 4.9±0.4 mM glucose (*Figure 1i* and *Figure 1—figure supplement 2*). Grouped data from six mice demonstrated that AVP neuron activity was consistently

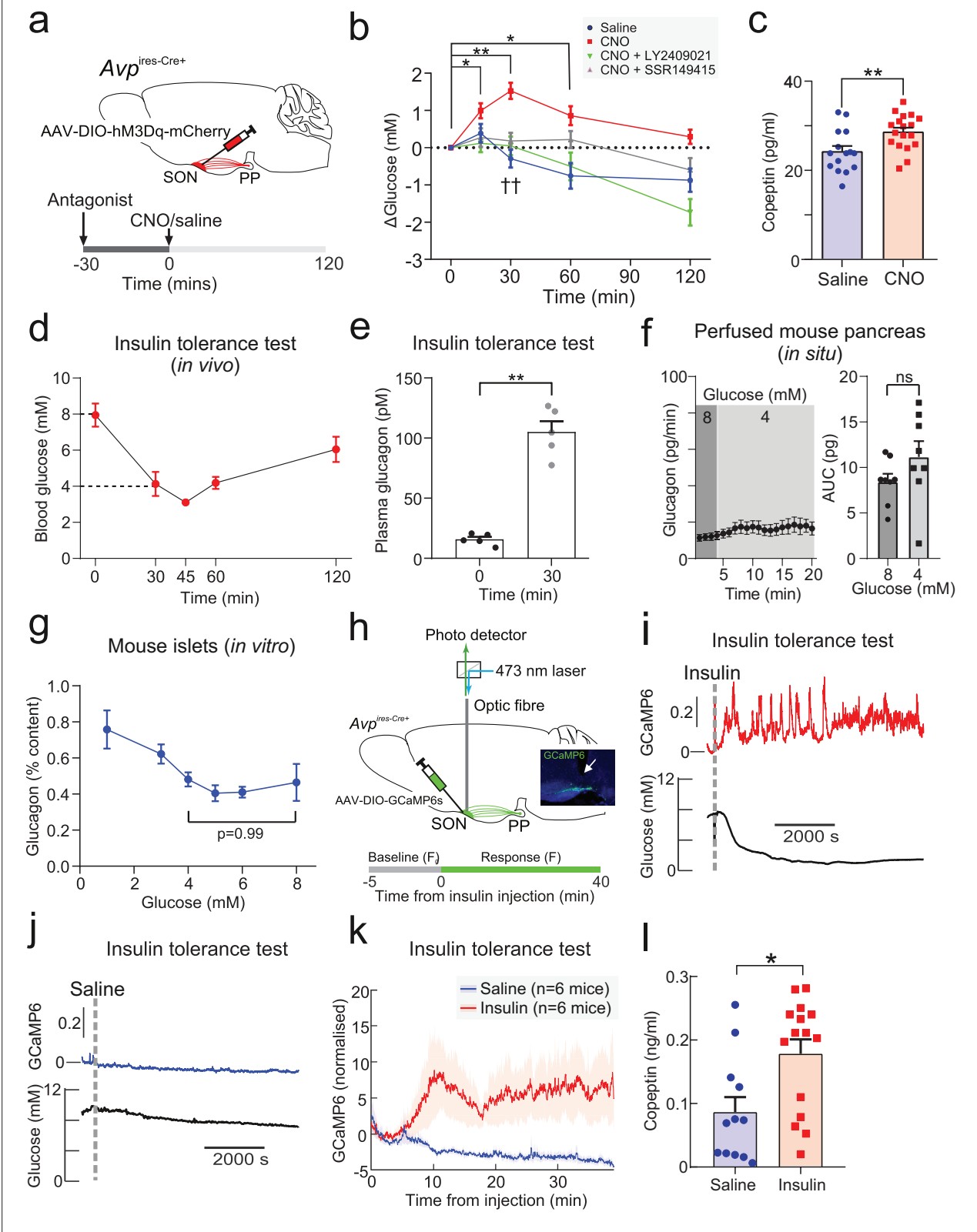

**Figure 1.** Insulin-induced hypoglycemia enhances population activity of AVP neurons in the supraoptic nucleus (SON), driving glucagon secretion AAV-DIO-hM3Dq-mCherry was injected bilaterally into the SON of *Avp*[ires-Cre/+] mice. CNO (3 mg/kg) or vehicle was injected i.p. In the same cohort (different trial), LY2409021 (5 mg/kg) or SSR149415 (30 mg/kg) was injected (i.p.) 30 min prior to CNO. See ***Figure 1—figure supplement 1***. (**a**) Blood glucose measurements from (**a**). Two-way RM ANOVA (Tukey's). CNO 0 min versus CNO at 15, 30, and 60 min; p<0.05=*, p<0.01=**. Comparison of

*Figure 1 continued on next page*

*Figure 1 continued*

CNO versus Saline, CNO+LY2409021 or CNO+SSR149415 at 30 min; p<0.01=††. Time, p<0.0001; Treatment, p=0.0006; Interaction, p<0.0001. n=6 mice. (**b**) Terminal plasma copeptin 30 min following saline or CNO injection. Mann-Whitney t-test (p=0.0025, **). n=15–18 mice. (**c**) Plasma glucose during an insulin tolerance test (ITT; 0.75 U/kg) in n=5 wild-type mice. (**d**) Plasma glucagon following an ITT. n=5 wild-type mice. Paired t-test, p<0.01=**. (**e**) Glucagon secretion from the perfused mouse pancreas. Glucagon released during all time points in 8 mM glucose was not significantly different from 4 mM glucose (all p>0.8). Right: area under curve. Paired t-test, ns=not significant. (**f**) Glucagon secretion from isolated mouse islets during 60 min static incubation at indicated glucose concentrations. n=7 wild-type mice. One-way ANOVA with Tukey post-hoc. 4 mM versus 8 mM glucose, p=0.99. (**g**) Measurements of population GCaMP6s activity in pituitary-projecting AVP neurons in the SON. Inset: Expression of GCaMP6s in AVP neurons in the SON. Arrow=tip of the optic fiber. (**h**) Simultaneous in vivo fiber photometry of AVP neuron activity (GCaMP6) and continuous glucose monitoring (black line) in response to an ITT (1 U/kg). Dashed gray line indicates the time of insulin injection. (**i**) Same animal as in (**j**), but for saline vehicle injection (dashed gray line). (**j**) GCaMP6s signal (normalized) in response to insulin (n=6 mice) or saline vehicle (n=6). Plasma copeptin at 30 min following saline or insulin. Mann-Whitney U-test, p=0.021. n=15–18 mice. AVP, arginine-vasopressin.

The online version of this article includes the following figure supplement(s) for figure 1:

**Figure supplement 1.** Effects of CNO in animals expressing mCherry in supraoptic nucleus (SON) AVP neurons (**a**) AAV-DIO-hM3Dq-mCherry was injected into the SON of $Avp^{ires-Cre/+}$ mice.

**Figure supplement 2.** Simultaneous continuous glucose monitoring (CGM) and in vivo fiber photometry of AVP neurons Grouped analysis of the glucose value at which the GCaMP6 signal crosses >2 SD from baseline, >3 SD from baseline and first exhibits a peak.

increased following insulin injection (*Figure 1k*). In contrast, saline vehicle did not increase AVP neuron activity (*Figure 1j and k*). We measured plasma copeptin in response to insulin-induced hypoglycemia. Again, these experiments were terminal due to copeptin sample volume requirements. In these experiments, copeptin was increased by 114% (*Figure 1l*).

## AVP stimulates glucagon-secreting pancreatic alpha-cells

To understand how AVP increases glucagon secretion, we characterized its effect on isolated islets. Mouse islets express mRNA for the vasopressin 1b receptor (V1bR; encoded by *Avpr1b*), whereas vasopressin receptor subtypes 1a and 2 mRNA (*Avpr1a* and *Avpr2*) were found in the heart and the kidneys, consistent with their distinct roles in the regulation of blood pressure and diuresis (*Bourque, 2008*), respectively (*Figure 2—figure supplement 1a*). To confirm that *Avpr1b* expression was enriched in alpha-cells, mice bearing a proglucagon promoter-driven Cre-recombinase (*Gcg*^Cre/+ mice) were crossed with mice expressing a Cre-driven fluorescent reporter (RFP). qPCR of the fluorescence-activated cell sorted RFP^+ and RFP^− fractions revealed that expression of *Avpr1b* is high in alpha-cells (RFP+) with ~43-fold enrichment above that seen in RFP-cells (principally beta-cells) (*Figure 2—figure supplement 1b*,c).

We explored whether the hyperglycemic and hyperglucagonemic actions of AVP are due to AVP directly stimulating glucagon secretion from alpha-cells. In dynamic measurements using the in situ perfused mouse pancreas, AVP produced a biphasic stimulation of glucagon secretion (*Figure 2a*). Glucagon secretion is a $Ca^{2+}$-dependent process (*Barg et al., 2000*). We therefore crossed *Gcg*^Cre/+ mice with a Cre-dependent GCaMP3 reporter mouse (from hereon, *Gcg*-GCaMP3 mice), and implanted islets from these mice in the anterior chamber of the eye of recipient wild-type mice (*Figure 2b–d*; see *Salem et al., 2019*). This allowed the cytoplasmic $Ca^{2+}$ concentration ($[Ca^{2+}]_i$) in individual alpha-cells to be imaged in vivo. Administration (i.v.) of AVP resulted in a biphasic elevation of $[Ca^{2+}]_i$ consisting of an initial spike followed by rapid oscillatory activity (*Figure 2c and d*), similar to the biphasic stimulation of glucagon secretion seen in the perfused pancreas.

In isolated mouse islets, AVP stimulated glucagon secretion (EC$_{50}$=25.0 pM; 95% confidence interval [CI]=[4.80, 133] pM) in islets incubated in 3 mM glucose (*Figure 2e*). The AVP-induced increase in glucagon was prevented by SSR149415 (*Figure 2f*). AVP also reversed the glucagonostatic effect of 15 mM glucose but higher concentrations (>1 nM) were required (*Figure 2—figure supplement 2a*). AVP at concentrations up to 100 nM did not affect insulin secretion from mouse islets when tested at 3 mM or 15 mM glucose (*Figure 2—figure supplement 2b*,c). Finally, AVP also stimulated glucagon secretion (EC$_{50}$=7.69 pM; 95% CI=[5.10, 113] pM) in isolated human pancreatic islets (*Figure 2g*).

To understand the intracellular mechanisms by which AVP stimulates glucagon secretion, we isolated islets from *Gcg*-GCaMP3 mice. We first performed perforated patch-clamp recordings of membrane potential in intact islets from *Gcg*-GCaMP3 mice. AVP increased action potential firing frequency (*Figure 3a and b*). Next, we conducted confocal imaging of $[Ca^{2+}]_i$ in these islets and confirmed

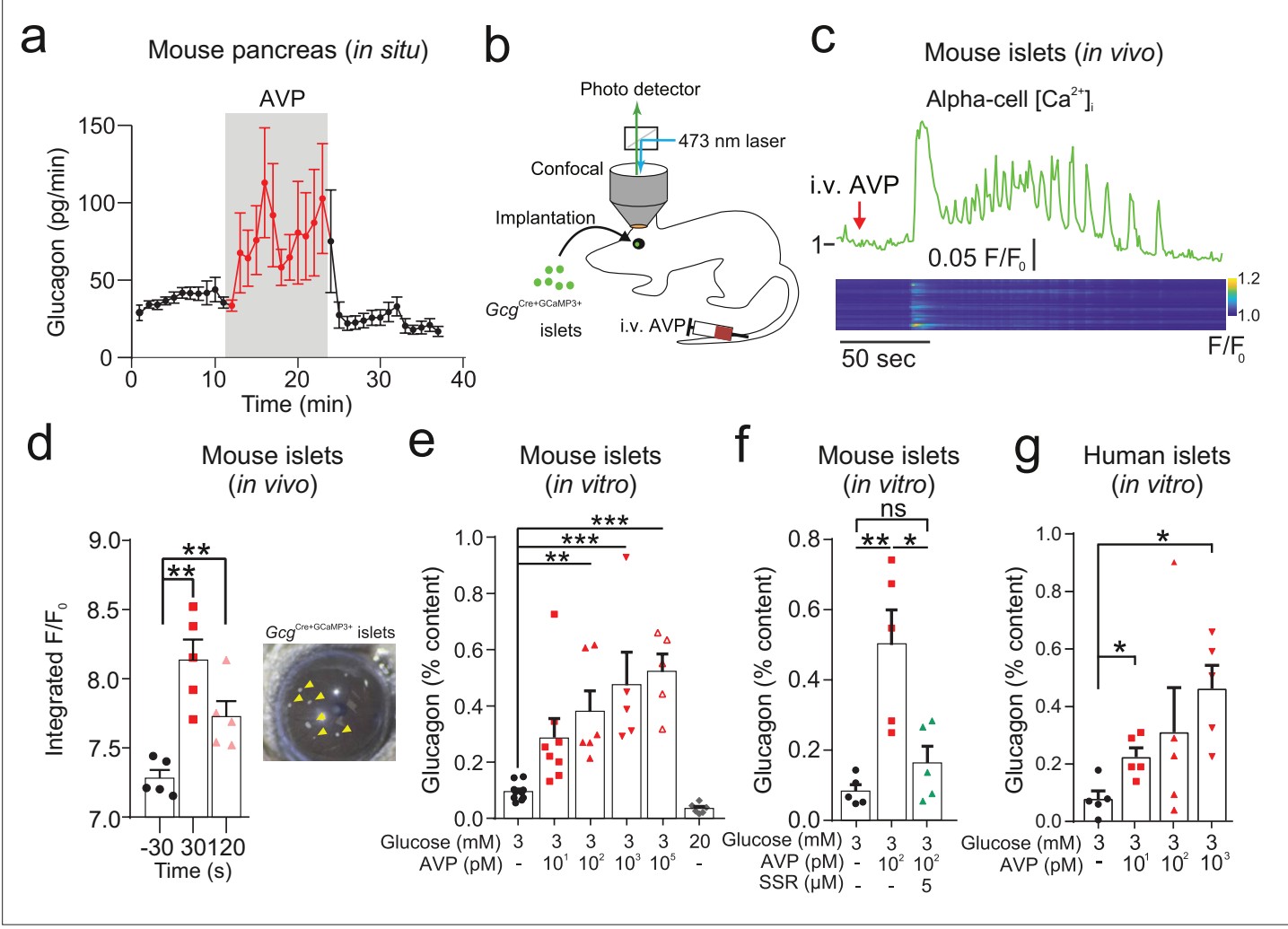

**Figure 2.** AVP increases glucagon release and alpha-cell activity ex vivo, in situ, and in vivo. (**a**) Glucagon secretion from the perfused mouse pancreas in response AVP (10 nM). All data are represented as mean ± SEM. n=5 mice. Extracellular glucose of 3 mM. (**b**) Islets from $Gcg^{Cre/+:GCaMP3/+}$ mice were injected into the anterior chamber of the eye (ACE) of recipient mice (n=5 islets in 5 mice). After >4 weeks, GCaMP3 was imaged in vivo in response to i.v. AVP (10 μg/kg) or saline administration. Saline did not change the GCaMP3 signal. Signal is GCaMP3 fluorescence (F) divided by baseline signal (F0). AVP evoked an increase in calcium activity, typically starting with a large transient. Below: Raster plot of response (normalized F/F0) in different cells (ROIs) with a single islet. (**c**) Response of alpha-cell to i.v. AVP. Lower panel shows raster plot of response in different cells. (**d**) Integrated F/F0 (area under curve) response for all alpha-cells in recorded islets (five islets, N=3 mice). The area under the curve was calculated 30 sec before i.v. injection, 30 sec after and 120 sec after. One-way RM ANOVA with Tukey's multiple comparison test; p<0.01=**. Right: Image of islets (arrows) engrafted in the ACE. (**e**) Glucagon secretion from isolated mouse islets in response to AVP. One-way ANOVA (p<0.05=*; p<0.01=**; p<0.001=***). n=5–10 wild-type mice in each condition. (**f**) Glucagon secretion from isolated mouse islets in response to AVP in the presence and absence of the V1bR antagonist SSR149415. One-way ANOVA (p<0.05=*; p<0.01=**). n=5 wild-type mice per condition. (**g**) Glucagon secretion from islets isolated from human donors, in response to AVP. Paired t-tests, p<0.05=*. n=5 human donors. AVP, arginine-vasopressin; ROIs, regions of interest.

The online version of this article includes the following figure supplement(s) for figure 2:

**Figure supplement 1.** Expression of the vasopressin 1b receptor in mouse and human.

**Figure supplement 2.** AVP and beta-cell function.

that AVP increased [Ca²⁺]ᵢ in alpha-cells (*Figure 3c and d*). The capacity of AVP to increase [Ca²⁺]ᵢ in alpha-cells was abolished following application of the V1bR antagonist SSR149415 (*Figure 3e*). AVP failed to stimulate Ca²⁺ activity in the presence of the Gq-inhibitor YM254890 (*Takasaki et al., 2004*; *Figure 3f*). AVP increased intracellular diacylglycerol, which is a downstream product of Gq activation (*Figure 3g and h*).

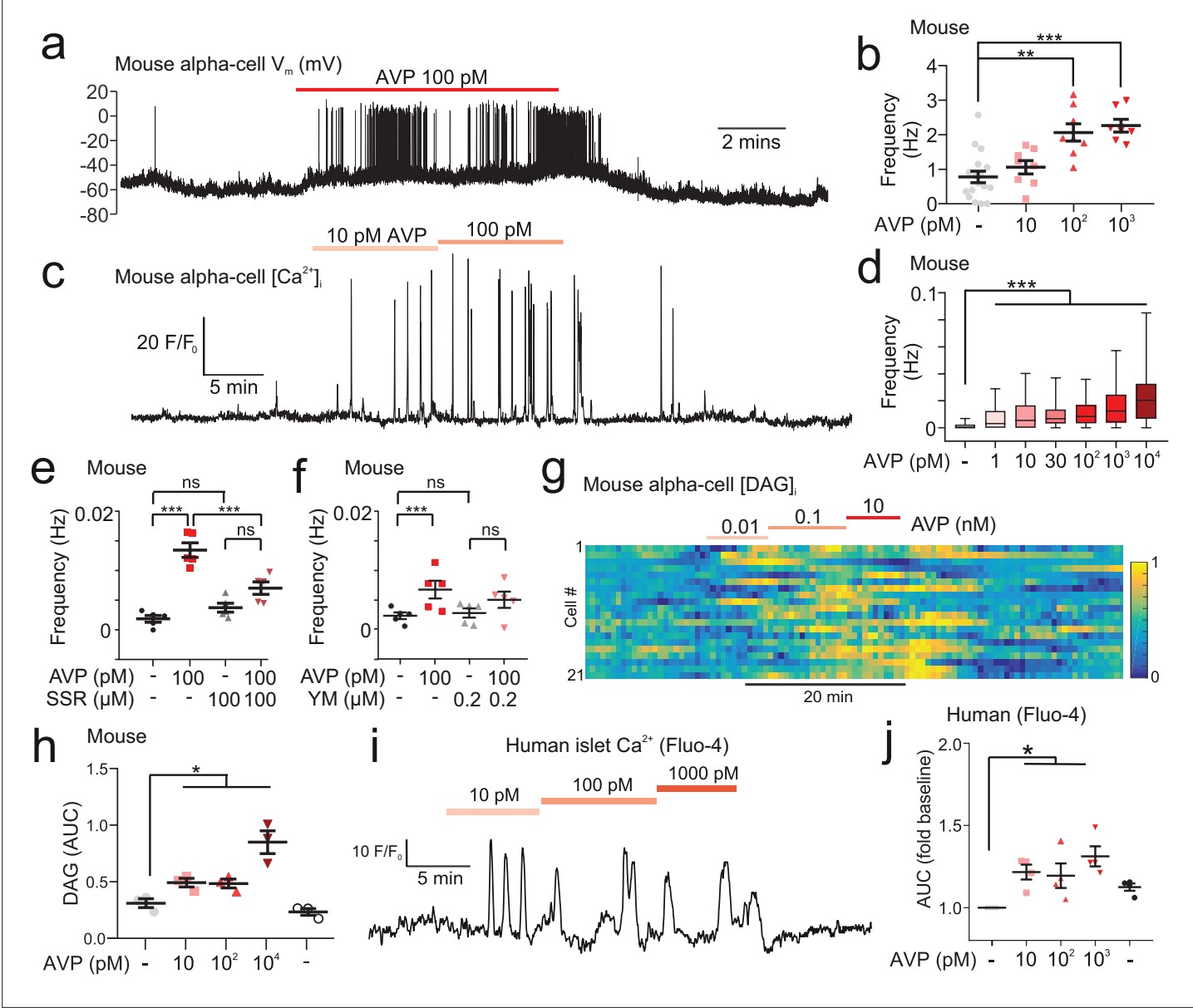

**Figure 3.** AVP increases action potential firing, Ca$^{2+}$ activity, and intracellular DAG in alpha-cells in intact islets. (**a**) Membrane potential (V$_m$) recording (perforated patch-clamp) of an alpha-cell in response to 100 pM AVP. (**b**) Frequency-response curve for varying concentrations of AVP (17 alpha-cells, 10 *Gcg*-GCaMP3 mice). Mixed-effects analysis of variance, Holm-Sidak's post-hoc (p<0.01=**; p<0.001=***; p=0.073 for 3 mM glucose vs. 10 pM AVP). (**c**) GCaMP3 signal from an alpha-cell in response to AVP. (**d**) Box and whisker plot of the frequency of GCaMP3 oscillations in response to AVP. 142–170 alpha-cells, 7 islets, n=7 *Gcg*-GCaMP3 mice. Recordings in 3 mM glucose. One-way RM ANOVA, p<0.001=***. (**e**) Frequency of GCaMP3 oscillations in response to 100 pM AVP in the presence and absence of SSR149415 (10 μM). 75–90 alpha-cells, 6 islets, n=5 *Gcg*-GCaMP3 mice. Recordings in 3 mM glucose. One-way ANOVA (Tukey), p<0.001=***, ns=not significant (p>0.2). (**f**) Frequency of GCaMP3 oscillations in response to 100 pM AVP in the absence and presence of YM-254890 (0.2 μM). 75–90 alpha-cells, 6 islets, n=5 *Gcg*$^{Cre/+;GCaMP3/+}$ mice. Recordings in 3 mM glucose. One-way RM ANOVA (Tukey's multiple comparisons test), p<0.05=*, ns=not significant (p>0.3). AVP versus AVP+YM-254890; p=0.99. (**g**) Heatmap of intracellular diacylglycerol (DAG; upward DAG) signal from single islet cells (dispersed into clusters) in response to AVP. The signal was median filtered and normalized to largest signal in the recording. (**h**) Area under curve (AUC, normalized to duration) for DAG signal for each experimental condition. 10 recordings, 152 cells, n=3 wild-type mice. One-way RM ANOVA, p<0.05=* (Tukey's multiple comparisons test). (**i**) Fluo4 signal from a putative alpha-cell in a human islet in response to AVP (10, 100, and 1000 pM). Recording in 3 mM glucose. (**j**) Area under curve (AUC, normalized to duration) for Fluo4 signal in each human islet, for each experimental condition. 26 islets, n=4 human donors. One-way ANOVA, p<0.05=* (Sidak). AVP, arginine-vasopressin.

In human islets, *AVPR1B* was the most abundant of the vasopressin receptor family (*Figure 2—figure supplement 1d*). These data are supported by bulk sequencing of human islet fractions (*Nica et al., 2013*; *Blodgett et al., 2015*). Finally, in alpha-cells in human islets (identified by their response to adrenaline *Hamilton et al., 2018*), AVP increased $[Ca^{2+}]_i$ (*Figure 3i and j*).

## Pharmacological and genetic inhibition of AVP signaling suppress counter-regulatory glucagon secretion

Given the relatively small increase in circulating copeptin in response to hypoglycemia (*Figure 1l*), we used pharmacological and genetic approaches to more conclusively establish the link between AVP and counter-regulatory glucagon during hypoglycemia. We injected wild-type mice with the V1bR antagonist SSR149415 prior to an insulin tolerance test (ITT) (*Figure 4a and b*). This reduced glucagon secretion during insulin-induced hypoglycemia by 60%. Similarly, in *Avpr1b* knockout mice (*Avpr1b$^{-/-}$*; *Lolait et al., 2007*) glucagon secretion was decreased by 65% compared to wild-type littermates (*Avpr1b$^{+/+}$*; *Figure 4c and d*). Despite the drastic reduction in plasma glucagon by pharmacological antagonism of the V1bR or genetic KO of *Avpr1b*, the depth of hypoglycemia was not affected (*Figure 4a and c*). Insulin measured during an ITT revealed that circulating insulin increases from a basal of 110±12 pM to 910±74 pM at 30 min (n=5 mice). Thus, insulin is present at an ~10-fold molar excess compared to glucagon. This explains why the hypoglycemic effect of exogenous insulin predominates in this experimental paradigm, with no change in plasma glucose despite a strong reduction in circulating glucagon. Indeed, in mice, an ITT tests both counter-regulation and insulin sensitivity as recently reviewed (*Virtue and Vidal-Puig, 2021*).

We also tested the ability of AVP to modulate glucagon in vivo in response to 2-deoxy-D-glucose (2DG). 2DG is a non-metabolizable glucose molecule that evokes a state of perceived glucose deficit (mimicking hypoglycemia) and triggers a robust counter-regulatory stimulation of glucagon secretion (*Marty et al., 2005*). We monitored AVP neuron activity in response to 2DG by in vivo fiber photometry, and correlated this to changes in plasma glucose during this metabolic challenge. Injection (i.p.) of 2DG increased blood glucose (*Figure 4e*) and triggered a concomitant elevation of $[Ca^{2+}]_i$ in AVP neurons (*Figure 4e–h*). The elevation in plasma glucagon by 2DG injection was attenuated by 50% following pretreatment with the V1bR antagonist SSR149415 (*Figure 4g*). The hyperglycemic response to 2DG was also partially antagonized by pretreatment with either the V1bR antagonist SSR149415 or the glucagon receptor antagonist LY2409021 (*Figure 4h and i*). We conclude that AVP contributes to the hyperglycemic response to 2DG, and it does so (at least in part) by stimulating glucagon release.

## Hypoglycemia evokes AVP secretion via activation of A1/C1 neurons

Many physiological stressors activate hindbrain catecholamine neurons, which release noradrenaline (A1) or adrenaline (C1) and reside in the ventrolateral portion of the medulla oblongata (VLM). Activation of C1 neurons (by targeted glucoprivation or chemogenetic manipulation) elevates blood glucose (*Ritter et al., 2000*; *Zhao et al., 2017*; *Li et al., 2018*) and plasma glucagon (*Andrew et al., 2007*). Furthermore, C1 cell lesions severely attenuate the release of AVP in response to hydralazine-evoked hypotension (*Madden et al., 2006*), indicating that this hindbrain site may be a key regulator of AVP neuron activity during physiological stress.

To characterize any functional connectivity between A1/C1 neurons and SON AVP neurons, we conducted channelrhodpsin-2-assisted circuit mapping (CRACM; *Petreanu et al., 2007*) and viral tracer studies (*Figure 5a–e*). We injected a Cre-dependent viral vector containing the light-gated ion channel channelrhodopsin-2 (AAV-DIO-ChR2-mCherry) into the VLM (targeting A1/C1 neurons) of *Avp$^{GFP/+}$::Th$^{Cre/+}$* mice (*Figure 5a*). Projections from the A1/C1 neurons were present in the SON and PVH and co-localized with AVP-immunoreactive neurons (*Figure 5b* and *Figure 5—figure supplement 1a*). A1/C1 neurons express vesicular glutamate transporter two and as a result co-release glutamate with catecholamines (see review *Guyenet et al., 2013*). Therefore, by monitoring glutamatergic excitatory post-synaptic currents (EPSCs) evoked by ChR2 activation, we could determine whether A1/C1 neurons are synaptically connected to AVP neurons. Brain slice electrophysiology revealed that in the majority (89%) of GFP$^+$ AVP neurons, opto-activation of A1/C1 neuron terminals results in EPSCs (*Figure 5a–e*). These EPSCs were glutamatergic, as they were abolished by the AMPA and kainate receptor antagonist DNQX (*Figure 5c*). Furthermore, these EPSCs could be blocked with TTX, but

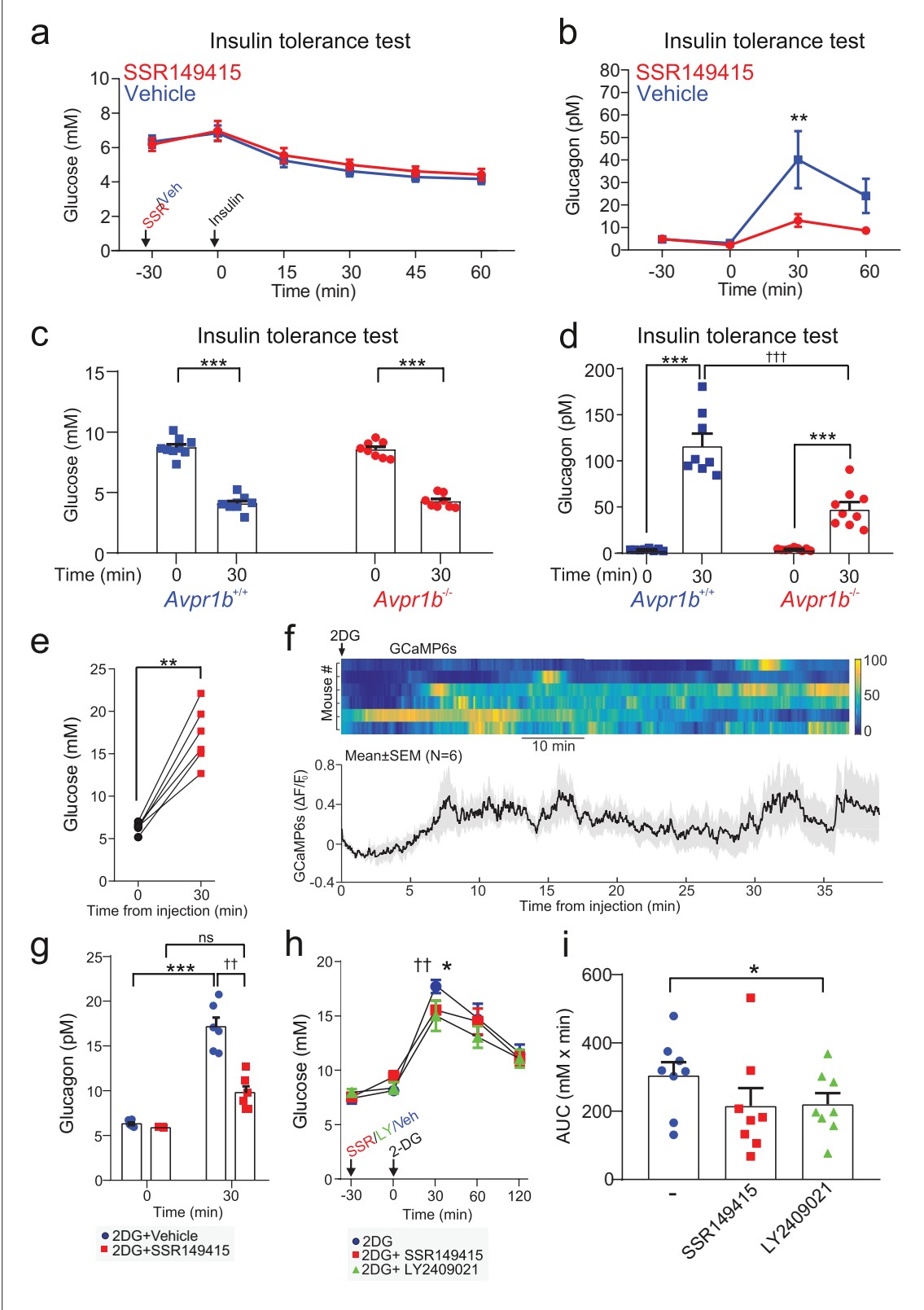

**Figure 4.** The vasopressin 1b receptor mediates hypoglycemia-induced glucagon secretion. (**a**) Blood glucose during an ITT (0.75 U/kg; injection at 30 min). 30 min prior to the commencement of the ITT (t=0 min), either the V1bR antagonist SSR149415 (30 mg/kg) or vehicle was administered i.p. n=7 wild-type mice. (**b**) Plasma glucagon for (**a**). Two-way RM ANOVA with Sidak's (between conditions) and Tukey's (within condition) multiple comparisons test. Vehicle versus SSR149415; p<0.01=†† (30 min) and p<0.05=† (60 min). Glucagon was increased in response to an ITT in both treatment groups

*Figure 4 continued on next page*

*Figure 4 continued*

(p<0.01=** vs. 0 min). n=6–7 wild-type mice. (**c**) Plasma glucose during an ITT (0.75 U/kg; injection at 0 min) in *Avpr1b*$^{-/-}$ mice and littermate controls (*Avpr1b*$^{+/+}$). Two-way RM ANOVA (Tukey), p<0.001=***. n=8–9 mice. (**d**) Plasma glucagon for (**c**). Two-way RM ANOVA (Sidak). *Avpr1b*$^{-/-}$ versus *Avpr1b*$^{+/+}$; p<0.001=††† (30 min). 0 min versus 30 min; p<0.001=***. n=8–9 mice. (**e**) Population GCaMP6s activity in pituitary-projecting AVP neurons in the supraoptic nucleus (SON). GCaMP6s was imaged in response to 2-Deoxy-D-glucose (2DG, 500 mg/kg) injection (i.p.). Plasma glucose at baseline (0 min) and 30 min after 2DG injection. Paired t-test, p<0.01=**. n=6 mice. (**f**) Upper: heatmap of population activity (GCaMP6s) response to 2DG for each mouse (n=6). Lower: mean ± SEM GCaMP6s signal for all mice (n=6) in response to 2DG. GCaMP6s data represented as $(F–F_0)/F_0$. (**g**) Plasma glucagon following 2DG injection (at 0 min). Prior to 2DG, either SSR149415 or vehicle was administered i.p. Two-way RM ANOVA by both factors (Bonferroni). Vehicle versus SSR149415; p<0.01=†† (30 min); p>0.99 (0 min). 0 min versus 30 min; p=0.009 (Vehicle); p=0.093 (SSR149415). Time, p<0.0001; Treatment, p=0.002; Interaction, p=0.005. n=6 mice. (**h**) Blood glucose response to 2DG, with or without pretreatment with SSR149415 (30 mg/kg), LY2409021 (5 mg/kg), or saline vehicle. Antagonists/vehicle injected 30 min prior to 2DG. Two-way RM ANOVA with Sidak's multiple comparison test; 2DG versus 2DG+SSR149415, p=0.0103 (*); 2DG versus 2DG+LY024091, p=0.0047 (††). (**i**) Area under the (glucose) curve (AUC). One way ANOVA; 2DG versus 2DG+LY024091, p<0.05 (*). ITT, insulin tolerance test.

reinstated with addition of 4-AP (*Figure 5e*), indicating that A1/C1 neurons connect to AVP neurons in the SON monosynaptically.

To explore the consequences of A1/C1 activation in vivo, we injected AAV-DIO-hM3Dq-mCherry bilaterally into the VLM of *Th*$^{Cre/+}$ mice (*Figure 5f* and *Figure 5—figure supplement 1b*,c). Activation of A1/C1 neurons with CNO evoked a ~4.5 mM increase in plasma glucose (*Figure 5g*). Pretreatment with the glucagon receptor antagonist LY2409021 inhibited this response (*Figure 5g*). In line with this, plasma glucagon was increased following CNO application (*Figure 5h*). The hyperglycemic response was also dependent on functional V1bRs, because it was abolished following pretreatment with the V1bR antagonist SSR149415 (*Figure 5g*). CNO had no effect on blood glucose in *Th*$^{Cre/+}$ mice expressing mCherry in A1/C1 neurons (*Figure 5—figure supplement 1c*). Taken together, these data indicate that A1/C1 neuron activation evokes AVP release, which in turn stimulates glucagon secretion. We therefore hypothesized that hypoglycemia-induced AVP release is due to projections from A1/C1 neurons. In support of this hypothesis, c-Fos expression (a marker of neuronal activity) was increased in A1/C1 neurons following an insulin bolus (*Figure 5—figure supplement 2a, b*).

To determine the contribution of the A1/C1 region to AVP neuron activity during an ITT, we inhibited the A1/C1 region whilst monitoring AVP neuron activity. To this end, we expressed an inhibitory receptor (the modified human muscarinic M4 receptor hM4Di; *Armbruster et al., 2007*) in A1/C1 neurons (by injecting AAV-fDIO-hM4Di-mCherry into the VLM) and GCaMP6s in AVP neurons (by injecting AAV-DIO-GCaMP6s into the SON) of *Dbh*$^{flp/+}$::*Avp*$^{ires-Cre/+}$ mice (*Figure 5i–k* and *Figure 5—figure supplement 2c*). We then measured AVP neuron population $[Ca^{2+}]_i$ activity (with in vivo fiber photometry) and plasma glucagon following inhibition of A1/C1 neurons with CNO. AVP neuron population activity during an ITT was partially reduced by A1/C1 silencing compared to vehicle injection (*Figure 5j*). Injection of CNO did not produce a statistically significant increase in basal plasma glucagon compared to saline (8±0.3 pM vs. 9±1 pM, respectively). However, when insulin was subsequently injected, glucagon rose significantly less in the CNO treated group (twofold vs. fourfold; *Figure 5k*). Taken together, these data suggest that A1/C1 neurons contribute to AVP-dependent glucagon secretion during an ITT.

## Insulin-induced AVP secretion underlies counter-regulatory glucagon secretion in humans

We extended these observations to human physiology. Healthy volunteers were given a hypoglycemic clamp during one visit and a euglycemic clamp during another visit in a randomized order (*Figure 6*). In response to hypoglycemia (blood glucose of 2.8±0.1 mM, *Figure 6a*), plasma glucagon increased by >300% (*Figure 6b*). In contrast, during euglycemia (blood glucose of 5.1±0.1 mM) glucagon was stable (*Figure 6b*). Measurements of plasma AVP revealed that AVP rose during a hypoglycemic clamp (from a basal 2 pM to 10 pM) but did not change during a euglycemic clamp (*Figure 6c*). There was a highly significant (p<0.001) correlation between AVP and glucagon (*Figure 6d*). Copeptin is more widely measured than AVP in clinical practice (*Morgenthaler et al., 2006*) and we therefore also measured the levels of this peptide. Like AVP, copeptin increased during the hypoglycemic clamp (*Figure 6e and f*). We compared copeptin and AVP measured in the same samples. Copeptin and AVP were significantly correlated but with a non-zero y-intercept (*Figure 6g*). This has been observed

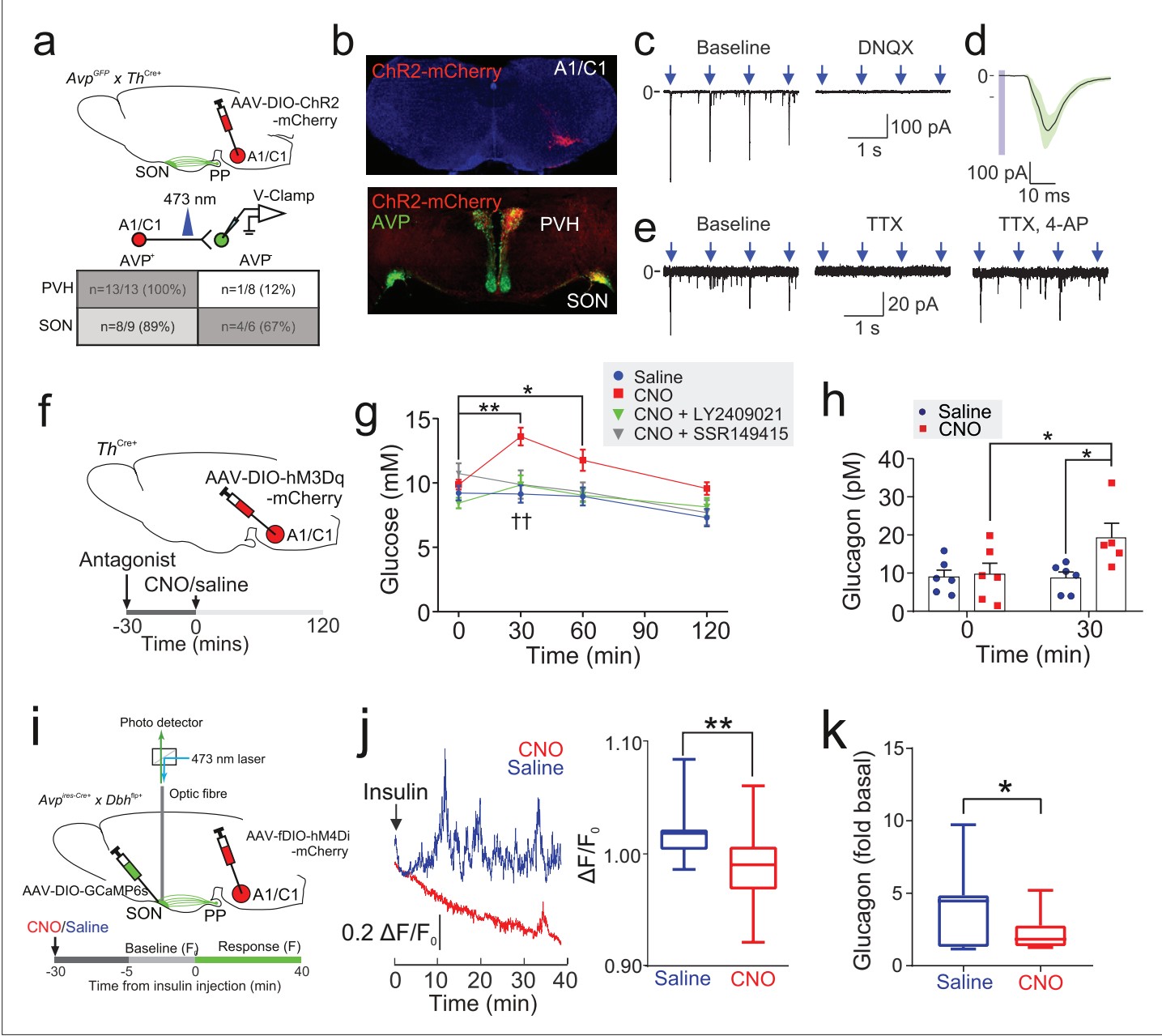

**Figure 5.** Insulin-induced AVP secretion is mediated by A1/C1 neurons. (**a**) Upper: AAV-DIO-ChR2-mCherry was injected into the VLM of $Th^{Cre/+}$ mice or $Avp^{GFP/+}::Th^{Cre/+}$ mice, targeting A1/C1 neurons. Lower: CRACM. Excitatory post-synaptic currents (EPSCs) were recorded in voltage-clamp mode in GFP$^+$ (AVP) and GFP$^-$ neurons in the SON and PVH. The number (n) of neurons that responded to opto-activating A1/C1 terminals. Total of 36 neurons recorded from five mice. See *Figure 5—figure supplement 1a*. (**b**) Upper: Viral expression of ChR2-mCherry in A1/C1 neurons. Lower: A1/C1 neuron terminals co-localize with AVP-immunoreactive neurons. Representative from three mice. (**c**) Left: EPSCs evoked by opto-activation of A1/C1 terminals with 473 nm light pulses (arrows). Right: Light-evoked EPSCs following application of DNQX (20 µM). Representative of five recordings from three mice. (**d**) EPSC waveforms in a single GFP$^+$ (AVP) neuron in response to repeated opto-activation of A1/C1 neuron terminals. Black line and shaded are=Mean ± SD of EPSCs. Light pulse=blue bar. Representative of three recordings from three mice. (**e**) EPSCs evoked by opto-activating A1/C1 terminals at baseline (left) and following addition of TTX (1 µM; middle) and 4-AP (1 mM; right). Representative of three recordings from three mice. (**f**) AAV-DIO-hM3Dq was injected into $Th^{Cre/+}$ mice, targeting A1/C1 neurons. CNO (1 mg/kg) was then injected (i.p.). Antagonists (or vehicle) for the V1bR (SSR149415, 30 mg/kg) or glucagon receptor (GCGR; LY2409021, 5 mg/kg) were injected 30 min prior to CNO. Plasma glucose and glucagon were then measured. See *Figure 5—figure supplement 1b,c*. (**g**) Plasma glucose in response to CNO and pretreatment with antagonists. n=8 mice. Two-way RM ANOVA (Sidak's multiple comparison's test). Time (p<0.0001), Treatment (p=0.03), and Interaction (p=0.0002). (**h**) Plasma glucagon at 30 min post-CNO (or vehicle) injection. Two-way RM ANOVA with Tukey's (within treatment) and Sidak's (between treatments) multiple comparisons. p<0.05=*, ns=not significant. Within treatment, CNO increased glucagon at 30 min versus 0 min (p=0.022). Saline did not (p=0.96). Between treatments, CNO increased

*Figure 5 continued on next page*

*Figure 5 continued*

glucagon at 30 min versus saline (p=0.001). n=6 mice. (**i**) In vivo fiber photometry measurements of population GCaMP6 activity in pituitary-projecting SON AVP neurons during A1/C1 neuron inhibition. AAV-DIO-GCaMP6s was injected into the SON and AAV-fDIO-hM4Di-mCherry into the VLM of *Avp*<sup>ires-Cre/+</sup>::*Dbh*<sup>flp/+</sup> mice. GCaMP6s was then imaged in response to an insulin tolerance test (ITT), following inhibition of the A1/C1 neuron (with CNO at 1 mg/kg), as indicated by the protocol in the lower horizontal bar. See *Figure 5—figure supplement 2c*. (**j**) Left: Example population activity in one mouse (as described in (**i**)) in response to an ITT, following saline or CNO treatment (on different trials). CNO strongly inhibited the response to insulin. Right: Average GCaMP6 signal (($F–F_0$)/$F_0$) during response to insulin with either saline or CNO pretreatment (n=9 mice). CNO reduces the AVP GCaMP6 signal. t-test, p<0.01=**. (**k**) Plasma glucagon in response to an ITT in mice described in (**i**). 30 min before the insulin injection, either saline or CNO was given i.p. Glucagon is represented as fold of basal, where basal is 0 min (just prior to insulin) and the sample was taken at 30 min post-insulin. t-test, p=0.023 (*). n=8 mice. AVP, arginine-vasopressin; SON, supraoptic nucleus.

The online version of this article includes the following figure supplement(s) for figure 5:

**Figure supplement 1.** Viral tracing of A1/C1 terminals.

**Figure supplement 2.** c-Fos expression in A1/C1 neurons during an ITT.

previously and is attributed to slower clearance kinetics of copeptin (*Fenske et al., 2018*; *Roussel et al., 2014*). This is important as it suggests that using copeptin as a surrogate marker of AVP underestimates the true changes in AVP. Nevertheless, there was a linear and statistically significant correlation between glucagon and copeptin in the samples taken from both eu- and hypoglycemic clamps (*Figure 6h*).

## Insulin-induced copeptin secretion is diminished in T1D

We measured copeptin and glucagon during hypoglycemic clamps in subjects with T1D and non-diabetic, BMI- and age-matched 'control' individuals (*Table 1*). As expected, the T1D patients had higher plasma glucose levels than the healthy controls (*Figure 7a*). During the hypoglycemic clamp, blood glucose was reduced in both controls and T1D participants, with blood glucose converging at 2.8 mM after 60 min (*Figure 7a*). In control subjects, hypoglycemia evoked a ~17 pM increase in plasma glucagon within 60 min (*Figure 7b*). In contrast, hypoglycemia failed to increase circulating glucagon in all subjects with T1D, even at 60 min (*Figure 7b*). For display, time-dependent changes in copeptin after induction of hypoglycemia are shown after subtraction of basal copeptin (7.0±0.8 pM and 4.5±0.6 pM in control and T1D participants, respectively). In line with the glucagon phenotype, copeptin was significantly elevated in controls but not in subjects with T1D (*Figure 7c*). We plotted plasma glucagon against total plasma copeptin. Overall, copeptin levels in the subjects with T1D clustered at the lower end of the relationship (*Figure 7d*).

## Discussion

We investigated the role of AVP in regulating glucagon secretion in vivo during hypoglycemia. In mouse, we first stimulated AVP neurons with a DREADD approach, and observed an increase in plasma glucose that was V1bR- and glucagon receptor-dependent. Administering AVP in vivo produced increases in circulating glucose and glucagon, and initiated [$Ca^{2+}$]$_i$ oscillations in alpha-cells transplanted into the anterior chamber of the eye. We also demonstrate that AVP neuron activity is elevated during hypoglycemia induced by either 2DG or exogenous insulin. Pharmacological antagonism or genetic KO of the V1bR revealed that AVP is a major contributor to counter-regulatory glucagon release.

We did not observe an effect of AVP on insulin secretion from isolated mouse islets, whether at low or high glucose. In contrast, earlier studies have demonstrated that AVP stimulates insulin secretion (*Dunning et al., 1984*; *Gao et al., 1992*; *Gao et al., 1990*; *Li et al., 1992*). We attribute this discrepancy to the much higher (un-physiological) concentrations used in the earlier studies, which might have resulted in off-target effects. Indeed, 10 nM AVP directly closes ATP-sensitive K<sup>+</sup> channels in insulin-secreting cell lines (*Martin et al., 1989*). It is notable that V1bR is the only receptor from the vasopressin receptor family expressed in mouse islets, and its expression is restricted to alpha-cells (*Taveau et al., 2017*; *DiGruccio et al., 2016*), an observation we now confirm. It is possible that the previous reports of stimulatory effects of AVP on insulin secretion might reflect paracrine stimulation mediated by glucagon (*Svendsen et al., 2018*). The fact that we did not observe such an amplification

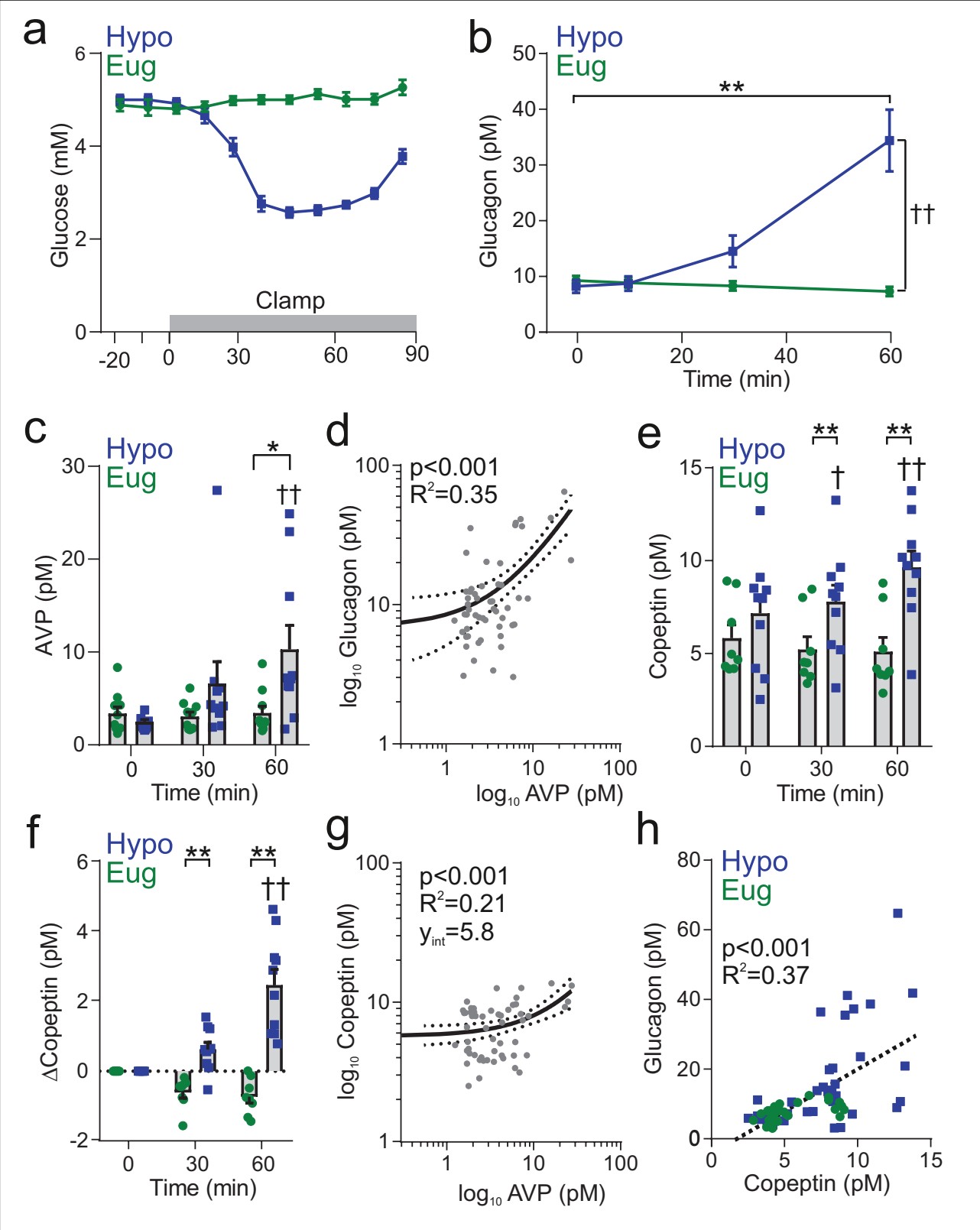

**Figure 6.** Insulin-induced hypoglycemia evokes copeptin and glucagon secretion in human participants. (**a**) Blood glucose was clamped at euglycemia (Eug) and followed during insulin-induced hypoglycemia (Hypo). n=10 healthy human subjects. The clamp was initiated at time 0 min and terminated at 60 min. (**b**) Plasma glucagon measurement during insulin-induced hypoglycemia. Two-way RM ANOVA by both factors. Time point versus 0 min; $p < 0.01 = **$. Between treatments; $p < 0.01 = ††$. (**c**) Plasma AVP measurement during and following clamping period. Two-way RM ANOVA by both factors.

*Figure 6 continued on next page*

Figure 6 continued

Hypoglycemia 0 min versus 60 min; p<0.01=††. Hypoglycemia 0 min versus 30 min; p=0.07. Between treatments; p<0.05=*. (**d**) Log-log plot of plasma AVP and plasma glucagon. Data points are 0, 30, and 60 min during hypoglycemic clamp. Linear regression (solid line) with 95% CI (dashed line). (**e**) Plasma copeptin measurement during hypoglycemic or euglycemic clamp. Two-way RM ANOVA by both factors. Indicated time point versus 0 min; p<0.05=†, p<0.01=††. For hypoglycemic clamp 0 min versus 30 min, p=0.07. Between treatments; p<0.01=**. (**f**) Change in plasma copeptin from baseline (time=0 min). Two-way RM ANOVA by both factors. Indicated time point versus 0 min; p<0.01=††. For hypoglycemic clamp 0 min versus 30 min, p=0.07. Between treatments; p<0.01=**. (**g**) Log-log plot of plasma AVP and plasma copeptin (as plotted in *Roussel et al., 2014*). Data points are 0, 30, and 60 min during hypoglycemic clamp. Linear regression (solid line) with 95% CI (dashed line). (**h**) Correlation of change in copeptin and glucagon, with a linear regression (dashed line). Data points are from both euglycemia and hypoglycemia at 0, 10, 30, and 60 min.

of insulin secretion by AVP may suggest that the stimulation of glucagon was not high enough to activate the GCGR or GLP1R in the beta-cells in the glucose concentrations tested.

In both mouse and human islets, we observed that AVP can stimulate glucagon release and increase alpha-cell $[Ca^{2+}]_i$. This is consistent with the expression of the vasopressin 1b receptor in alpha-cells. Interestingly, other receptors from the vasopressin receptor family (namely, the oxytocin receptor) were expressed in both mouse and human alpha-cells at similar levels. Additional experiments are required to determine the relative physiological importance of oxytocin and AVP in the control of glucagon secretion. We also acknowledge that it remains to be formally demonstrated (using V1bR antagonists) that AVP stimulates glucagon secretion in human islets via activation of the vasopressin 1b receptor.

Catecholaminergic neurons in the VLM are a key component of the central counter-regulatory circuit and the ability of these neurons to evoke hyperglycemia (*Ritter et al., 2000*; *Zhao et al., 2017*; *Li et al., 2018*) and hyperglucagonemia (*Andrew et al., 2007*) is well-established. We confirm that activation of A1/C1 neurons evokes hyperglycemia, but by promoting glucagon release. Recent studies have clearly demonstrated that spinally-projecting C1 neurons evoke hyperglycemia by stimulating the adrenal medulla (*Zhao et al., 2017*; *Li et al., 2018*), suggesting that corticosterone and/or adrenaline could be contributing to glucagon release in this setting. However, circulating (nanomolar) levels of adrenaline do not increase glucagon release from isolated islets (*De Marinis et al., 2010*). Nevertheless, it should be noted that we observed that V1bR blockade did not completely abolish insulin- and 2-DG-induced glucagon release, highlighting the involvement of other signals such as amino acids (*Holst et al., 2017*) and/or adrenaline (*Hamilton et al., 2018*) (the latter released locally in the islet from adrenergic nerve terminals). Neurons in the VLM have a well-documented ability to increase plasma AVP (*Madden et al., 2006*; *Ross et al., 1984*). Our data strongly support the notion that the hyperglycemic and hyperglucagonemic effect of activating A1/C1 neurons is, at least in part, mediated by stimulation of AVP release: we show that A1/C1 neurons send functional projections to the SON, A1/C1 function is required for insulin-induced AVP neuron activation and that AVP is an important stimulus of glucagon secretion. However, we recognize that other circuits must be involved in the activation of AVP neurons, because A1/C1 neuron inhibition only partially prevented insulin-induced AVP neuron activation and glucagon release (see *Figure 5j and k*; although this may also be explained by variability in AAV-DIO-GCaMP6 and AAV-fDIO-hM4Di transfection in these experiments). For example, the PVN is richly supplied with axons from the BNST (*Sawchenko and Swanson, 1983*) and hypothalamic VMN neurons are key drivers of the glucose counter-regulatory response project to the BNST (*Meek et al., 2016*). Therefore, a VHN-BNST circuit may also be important for driving AVP neuron activity in response to hypoglycemia, explaining why A1/C1 inhibition only partially prevented the activation of AVP neurons. AVP infusion in human participants increases circulating glucagon (*Spruce et al., 1985*), but it is also thought to directly stimulate glycogen breakdown from the liver

**Table 1.** Participant characteristics for *Figure 7*.

|  | Christensen et al., 2011 | Christensen et al., 2015 | P |
|---|---|---|---|
| Cohort | Control (n=10) | T1D (n=10) |  |
| Age (years) | 23±1 | 26±1 | 0.06 |
| BMI (kg/m²) | 23±0.5 | 24±0.5 | 0.1744 |
| HbA1c (%) | 5.5±0.1 | 7.3±0.2 | <0.0001 |

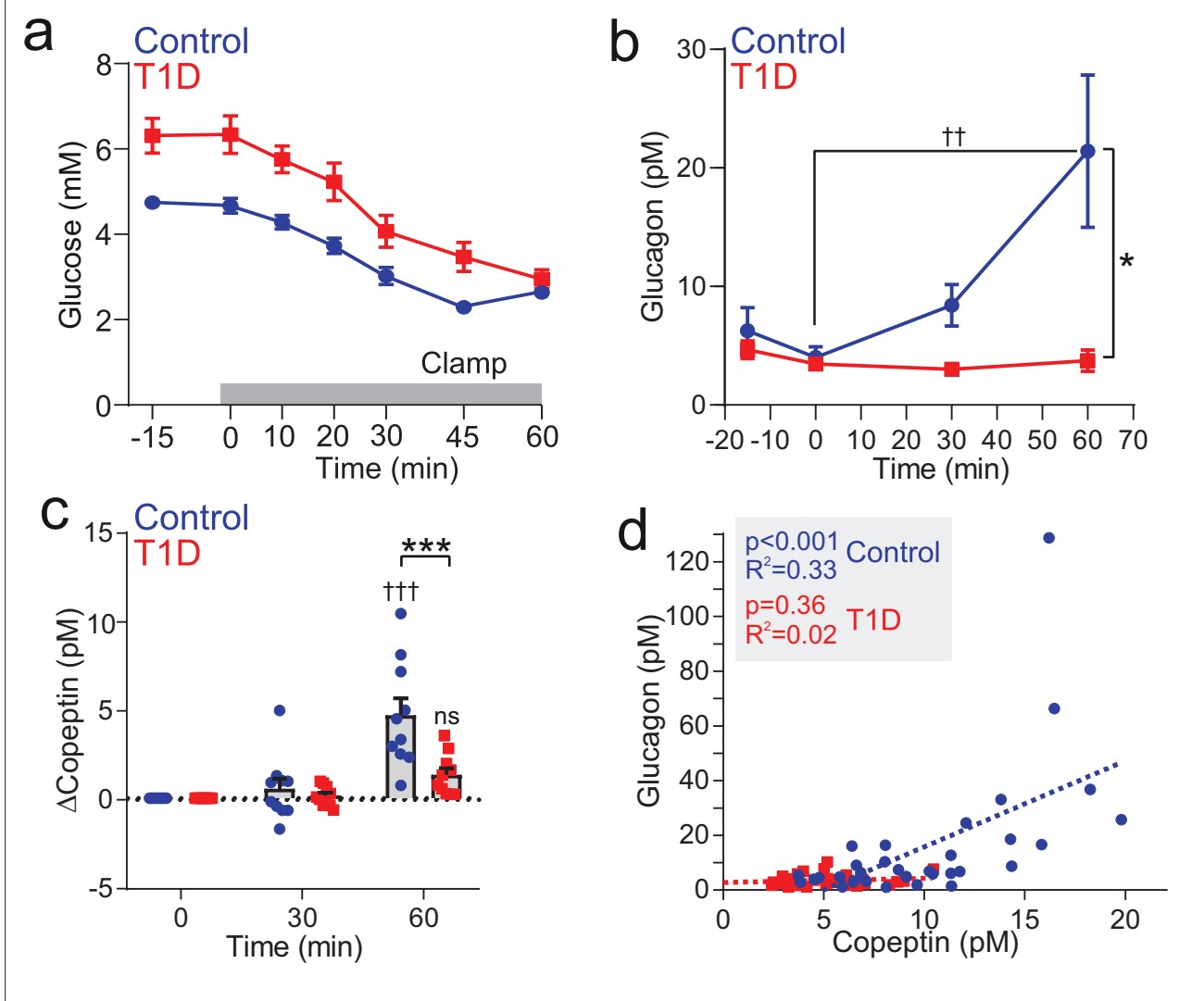

**Figure 7.** Insulin-induced copeptin and glucagon secretion is diminished in people with type 1 diabetes. (**a**) Hypoglycemia was induced by an insulin infusion in patients with T1D (n=10) and non-diabetic individuals (Controls; n=10). The insulin infusion was initiated at 0 min. (**b**) Plasma glucagon during the clamping period. Two-way RM ANOVA by both factors. Indicated time point versus 0 min; $p<0.01$=††. Between groups; $p<0.05$=*. Data from n=10 Control and n=10 T1D. (**c**) Plasma copeptin measurement during and following clamping period. Change in copeptin from baseline (time=0 min). Two-way RM ANOVA by both factors. Time point versus 0 min; $p<0.001$=†††; ns = not significant ($p>0.05$). Between groups; $p<0.001$=***. Data from n=10 Control and n=10 T1D. (**d**) Correlation of copeptin and glucagon following hypoglycemic clamping for control participants (n=10, circle) and T1D (n=10, square). Linear regressions (dashed lines) for Control and T1D data sets.

(***Hems et al., 1978***). Consequently, it is possible that part of the increase in plasma glucose following A1/C1 neuron stimulation is mediated by the action of AVP on the liver.

The signal driving AVP neuron activation is unlikely to be due to a direct action of insulin on AVP or A1/C1 neurons, because 2-DG caused a similar activation. The widely accepted view is that the activation of A1/C1 neurons (and consequently AVP neurons) during hypoglycemia depends on peripheral glucose sensing at multiple sites, including the hepatic portal system (***Verberne et al., 2014***; ***Marty et al., 2007***). We now show that the activity of AVP neurons is increased when plasma glucose fall to ~4.9 mM. The exact location of the glucose sensing is still controversial. It is unknown whether this threshold is sufficient to directly activate A1/C1 neurons. However, vagal afferents in the hepatic portal vein—which convey vagal sensory information to the A1/C1 neurons via projections to the nucleus of the solitary tract (***Verberne et al., 2014***)—are responsive to alterations in glucose at this threshold (***Niijima, 1982***).

In both mouse and human plasma, there was a relatively high basal level of copeptin. The high basal levels of copeptin likely relate to the longer half-life of this larger peptide (*Fenske et al., 2018*; *Roussel et al., 2014*). Plasma AVP undergoes much more dramatic variations (*Fenske et al., 2018*; *Roussel et al., 2014*), as our human data demonstrate but such measurements are currently not feasible in mice (*Morgenthaler et al., 2008*). We found that plasma AVP levels during hypoglycemia vary between 1 pM and 30 pM (*Figure 6c and d*), in good agreement with previous studies measuring AVP with a radioimmunoassay in human and rat (*Baylis and Robertson, 1980*; *Baylis and De Beer, 1981*). This range of concentrations is also in line with the glucagonotrophic effect of AVP in vitro (*Figure 2e and g*) and the ability of AVP to increase alpha-cell membrane potential, $[Ca^{2+}]_i$, and $[DAG]_i$ (*Figure 3*).

Cranial diabetes insipidus is a condition caused by reduced AVP production in the pituitary. Our findings provide an explanation to observations made 50 years ago that patients with cranial diabetes insipidus have a dramatically increased risk of hypoglycemia in response to the sulfonylurea chlorpropamide (*Ehrlich and Kooh, 1969*; *Vallet et al., 1970*; *Ehrlich and Kooh, 1970*), with hypoglycemia occurring with prevalence of 50% and therefore considered the major barrier of this treatment (*Ehrlich and Kooh, 1969*; *Vallet et al., 1970*; *Ehrlich and Kooh, 1970*). The present data raise the interesting possibility that this reflects the loss of AVP-induced glucagon secretion in these patients.

In T1D, deficiency of the secretory response of glucagon to hypoglycemia is an early acquired (<2 years of onset) abnormality of counter-regulation and leads to severe hypoglycemia (*Cryer, 2002*; *Siafarikas et al., 2012*), which may in part be explained by the major structural and functional changes that occur to islets in T1D. However, we also found that insulin-induced copeptin secretion was reduced, with some subjects exhibiting no elevation in copeptin. Blood glucose levels are higher in patients with T1D. It could be argued that the plasma copeptin levels are lower in T1D patients because they have not been hypoglycemic for a sufficiently long period. Although this possibility cannot fully be excluded, it seems unlikely given the strong relationship between plasma AVP/copeptin and glucagon secretion in non-diabetic individuals (*Figures 6d, g , and 7d*) and the finding that this correlation is not seen in patients with T1D (*Figure 7d*). Future studies with prolonged hypoglycemia in patients with T1D may help to conclusively resolve the matter. The reason for diminished hypoglycemia/insulin-induced copeptin release in T1D is unknown, but we speculate that recurrent hypoglycemia in patients with T1D may result in changes in glucose and/or insulin sensitivity in the A1/C1 region. Hypoglycemia awareness—a symptomatic component of which is driven by the autonomic nervous system—is associated with decreased insulin-induced copeptin in T1D patients (*Seelig et al., 2013*). Our data suggest that monitoring of copeptin may prove an important tool for stratification of hypoglycemia risk in patients with type 1 diabetes.

## Materials and methods

### Animals

All animals were kept in a specific pathogen-free (SPF) facility under a 12:12 hr light:dark cycle at 22°C, with unrestricted access to standard rodent chow and water. C57BL/6J mice used in this study are referred to as wild-type mice. To generate alpha-cell specific expression of the genetically encoded $Ca^{2+}$ sensor GCaMP3, mice carrying Cre recombinase under the control of the proglucagon promoter ($Gcg^{Cre/+}$ mice) were crossed with mice with a floxed green calmodulin (GCaMP3) $Ca^{2+}$ indicator in the ROSA26 locus (The Jackson Laboratory). These mice are referred to as *Gcg*-GCaMP3 mice. To generate mice expressing RFP in alpha-cells, $Gcg^{Cre/+}$ were crossed with mice containing a floxed tandem-dimer red fluorescent protein (tdRFP) in the ROSA26 locus (*Gcg*-RFP mice). Both of these mouse models were kept on a C57BL/6J background. Other transgenic mouse strains used— namely, $Avp^{ires-Cre/+}$ (*Pei et al., 2014*), $Th^{Cre/+}$ (The Jackson Laboratory), $Dbh^{flp/+}$ (MMRCC), and $Avp^{GFP/+}$ (MMRCC)—were heterozygous for the transgene and maintained on a mixed background. $Avpr1b^{-/-}$ and littermate controls ($Avpr1b^{+/+}$) were bred and maintained as previously described (*Wersinger et al., 2002*).

### Isolation of mouse islets

Mice of both sex and 11–16 weeks of age were killed by cervical dislocation (UK Schedule one procedure). Pancreatic islets were isolated by liberase digestion followed by manual picking. Islets were

used acutely and were, pending the experiments, maintained in tissue culture for <36 hr in RPMI 1640 (11879-020, Gibco, Thermo Fisher Scientific) containing 1% penicillin/streptomycin (1214-122, Gibco, Thermo Fisher Scientific), 10% fetal bovine serum (FBS) (F7524-500G, Sigma-Aldrich), and 11 mM glucose, prior to experiments.

## Patch-clamp electrophysiology in islets

Mouse islets were used for patch-clamp electrophysiological recordings. These recordings (in intact islets) were performed at 33–34°C using an EPC-10 patch-clamp amplifier and PatchMaster software (HEKA Electronics, Lambrecht/Pfalz, Germany). Unless otherwise stated, recordings were made in 3 mM glucose, to mimic hypoglycemic conditions in mice. Currents were filtered at 2.9 kHz and digitized at >10 kHz. A new islet was used for each recording. Membrane potential $(V_M)$ recordings were conducted using the perforated patch-clamp technique, as previously described (*Briant et al., 2018b*; *Kellard et al., 2020*). The pipette solution contained (in mM) 76 $K_2SO_4$, 10NaCl, 10 KCl, 1 $MgCl_2 \cdot 6H_2O$, and 5 Hepes (pH 7.35 with KOH). For these experiments, the bath solution contained (mM) 140 NaCl, 3.6 KCl, 10 Hepes, 0.5 $MgCl_2 \cdot 6H_2O$, 0.5 $Na_2H_2PO_4$, 5 $NaHCO_3$, and 1.5 $CaCl_2$ (pH 7.4 with NaOH). Amphotericin B (final concentration of 25 mg/ml, Sigma-Aldrich) was added to the pipette solution to give electrical access to the cells (series resistance of <100 MΩ). Alpha-cells in *Gcg*-GCaMP3 islets were confirmed by the presence of GCaMP3.

The frequency of action potential firing was calculated in MATLAB v.6.1 (2000; The MathWorks, Natick, MA). In brief, a peak-find algorithm was used to detect action potentials. This was then used to calculate firing frequency in different experimental conditions (AVP concentrations). Power-spectrum analysis of $V_M$ was conducted in Spike2 (CED, Cambridge, UK). $V_M$ was moving-average filtered (interval of 200 ms) and the mean $V_M$ subtracted. A power-spectrum was then produced (Hanning window with 0.15 Hz resolution) during 3 mM glucose alone, and with 10 pM AVP.

## GCaMP3 imaging in mouse islets

Time-lapse imaging of the intracellular $Ca^{2+}$ concentration ($[Ca^{2+}]_i$) in *Gcg*-GCaMP3 mouse islets was performed on an inverted Zeiss AxioVert 200 microscope, equipped with the Zeiss LSM 510-META laser confocal scanning system, using a 403/1.3 NA objective. Mouse islets were transferred into a custom-built recording chamber. Islets were then continuously perfused with bath solution at a rate of 200 µl/min. The bath solution contained (in mM): 140 NaCl, 5 KCl, 1.2 $MgCl_2$, 2.6 $CaCl_2$, 1 $NaH_2PO_4$, 10 Hepes, 17 mannitol, and 3 glucose. GCaMP3 was excited at 430 nm and recorded at 300–530 nm. The pinhole diameter was kept constant, and frames of 256×256 pixels were taken every 800 ms. Unless otherwise stated, recordings were made in 3 mM glucose, to mimic hypoglycemic conditions in mice. Raw GCaMP3 data was processed as follows; regions of interest (ROIs) were manually drawn around each GCaMP3+ cell in ImageJ and the time series of the GCaMP3 signal for each cell was exported. These data were first imported into Spike2 7.04 (CED, Cambridge, UK), wherein the data was median filtered to remove baseline drift. The size of the filter was optimized for each individual cell to remove drift/artifacts but preserve $Ca^{2+}$ transients. $Ca^{2+}$ transients were then automatically detected using the built in peak-find algorithm; the amplitude of peaks to be detected was dependent on the SNR but was typically >20 % of the maximal signal intensity. Following this, frequency of $Ca^{2+}$ transients could be determined. For plotting $Ca^{2+}$ data, the data was imported into MATLAB.

## DAG measurements in mouse islets

The effects of AVP on the intracellular diacylglycerol concentration (DAG) in pancreatic islet cells were studied using a recombinant circularly permutated probe, Upward DAG (Montana Molecular). Islets isolated from wild-type mice were gently dispersed (using Trypsin ES) into clusters and platted on rectangular coverslips. Cell clusters were then transfected with Upward DAG, delivered via a BacMam infection (according to the manufacturer's guidelines). Coverslips were then were placed in a custom-built chamber. Imaging experiments were performed 36–48 hr after infection using a Zeiss Axio-Zoom.V16 zoom microscope equipped with a 2.3×/0.57 objective (Carl Zeiss). The fluorescence was excited at 480 nm, and the emitted light was collected at 515 nm. The cells were kept at 33–35°C and were perfused continuously throughout the experiment with KRB solution supplemented with 3 mM glucose. The images were acquired using Zen Blue software (Carl Zeiss). The mean intensity for each cell was determined by manually drawing ROIs in ImageJ. Data analysis and representation were

performed with MATLAB. All data were processed using a moving average filter function (*smooth*) with a span of 50 min, minimum subtracted and then normalized to maximum signal intensity in the time series. AUC was calculated using the *trapz* function and then divided by the length of the condition.

### Ca²⁺ imaging in human islets

Time-lapse imaging of $[Ca^{2+}]_i$ in human islets was performed on the inverted Zeiss SteREO Discovery V20 Microscope, using a PlanApo S 3.5× mono objective. Human islets were loaded with 5 µg/µl of the $Ca^{2+}$-sensitive dye Fluo-4 (1923626, Invitrogen, Thermo Fisher Scientific) for 60 min before being transferred to a recording chamber. Islets were then continuously perfused with Dulbecco's modified Eagle's medium (DMEM) (11885-084, Gibco, Thermo Fisher Scientific) with 10% FBS, 100 units/ml penicillin, and 100 mg/ml streptomycin at a rate of 200 µl/min. Fluo-4 was excited at 488 nm and fluorescence emission was collected at 530 nm. The pinhole diameter was kept constant, and frames of 1388×1040 pixels were taken every 3 s. The mean intensity for each islet was determined by manually drawing an ROI around the islet in ImageJ. Data analysis and representation were performed with MATLAB. All data were processed using a moving average filter function (*smooth*) with a span of 20 min, minimum subtracted and then normalized to maximum signal intensity in the time series. AUC was calculated using the *trapz* function and then divided by the length of the condition.

### Pancreatic islet isolation, transplantation, and in vivo imaging of islets implanted into the anterior chamber of the eye

Pancreatic islets from *Gcg*-GCaMP3 mice were isolated and cultured as described above. For transplantation, 10–20 islets were aspirated with a 27-gauge blunt eye cannula (BeaverVisitec, UK) connected to a 100-µl Hamilton syringe (Hamilton) via 0.4 mm polyethylene tubing (Portex Limited). Prior to surgery, mice (C57BL6/J) were anesthetised with 2–4% isoflurane (Zoetis) and placed in a stereotactic frame. The cornea was incised near the junction with the sclera, then the blunt cannula (pre-loaded with islets) was inserted into the ACE and islets were expelled (average injection volume 20 µl for 10 islets). Carprofen (Bayer, UK) and eye ointment were administered post-surgery. A minimum of 4 weeks was allowed for full implantation before imaging. Imaging sessions were performed with the mouse held in a stereotactic frame and the eye gently retracted, with the animal maintained under 2–4% isoflurane anesthesia. All imaging experiments were conducted using a spinning disk confocal microscope (Nikon Eclipse Ti, Crest spinning disk, 20× water dipping 1.0 NA objective). The signal from GCaMP3 (ex. 488 nm, em. 525±25 nm) was monitored at 3 Hz for up to 20 min. After a baseline recording, mice received a bolus of AVP (10 µg/kg) i.v. (tail vein). Data were imported into ImageJ for initial movement correction (conducted with the StackReg and TurboReg plugins) and ROI selection. Analysis was then conducted in MATLAB.

### Hormone secretion measurements from mouse and human islets

Islets, from human donors or isolated from wild-type mice, were incubated for 1 hr in RPMI or DMEM supplemented with 7.5 mM glucose in a cell culture incubator. Size-matched batches of 15–20 islets were pre-incubated in 0.2 ml KRB with 2 mg/ml BSA (S6003, Sigma-Aldrich) and 3 mM glucose for 1 hr in a water-bath at 37°C. Following this islets were statically subjected to 0.2 ml KRB with 2 mg/ml BSA with the condition (e.g., 10 pM AVP) for 1 hr. After each incubation, the supernatant was removed and kept, and 0.1 ml of acid:etoh (1:15) was added to the islets. Both of these were then stored at –80°C. Each condition was repeated in at least triplicates.

Glucagon and insulin measurements in supernatants and content measurements were performed using a dual mouse insulin/glucagon assay system (Meso Scale Discovery, MD) according to the protocol provided.

### Hormone secretion measurements in the perfused mouse pancreas

Dynamic measurements of glucagon were performed using the in situ perfused mouse pancreas. Briefly, the aorta was cannulated by ligating above the coeliac artery and below the superior mesenteric artery, and the pancreas was perfused with KRB at a rate of ~0.45 ml/min using an Ismatec Reglo Digital MS2/12 peristaltic pump. The KRB solution was maintained at 37°C with a Warner Instruments temperature control unit TC-32 4B in conjunction with a tube heater (Warner Instruments P/N

64-0102) and a Harvard Apparatus heated rodent operating table. The effluent was collected by cannulating the portal vein and using a Teledyne ISCO Foxy R1 fraction collector. The pancreas was first perfused for 10 min with 3 mM glucose before commencing the experiment to establish the basal rate of secretion. Glucagon measurements in collected effluent were performed using RIA.

## Flow cytometry of islet cells (FACS), RNA extraction, cDNA synthesis, and quantitative PCR

The expression of the AVPR gene family was analyzed in tissues from 12-week old C57BL6/J mice ( three mice) and pancreatic islets from human donors ( two samples, each comprised of pooled islet cDNA from 7 and 8 donors, respectively). Total RNA was isolated using a combination of TRIzol and PureLink RNA Mini Kit (Ambion, Thermo Fisher Scientific) with incorporated DNase treatment.

Pancreatic islets from *Gcg*-RFP mice were isolated and then dissociated into single cells by trypsin digestion and mechanical dissociation. Single cells were passed through a MoFlo Legacy (Beckman Coulter). Cells were purified by combining several narrow gates. Forward and side scatter were used to isolate small cells and to exclude cell debris. Cells were then gated on pulse width to exclude doublets or triplets. RFP$^+$ cells were excited with a 488-nm laser and the fluorescent signal was detected through a 580/30 bandpass filter (i.e., in the range 565–595 nm). RFP-negative cells were collected in parallel. The levels of gene expression in the RFP$^+$ and in the RFP$^-$ FAC-sorted fractions were determined using real-time quantitative PCR (qPCR). RNA from FACS-sorted islet cells was isolated using RNeasy Micro Kit (Qiagen). cDNA was synthesized using the High Capacity RNA-to-cDNA Kit (Applied Biosystems, Thermo Fisher Scientific). Real-time qPCR was performed using SYBR Green detection and gene-specific QuantiTect Primer Assays (Qiagen) on a 7900HT Applied Biosystems analyzer. All reactions were run in triplicates. Relative expression was calculated using ΔCt method *Actb* as a reference gene.

## Stereotaxic surgery and viral injections

For viral injections into the SON, mice were anesthetized with ketamine/xylazine (100 mg/kg and 10 mg/kg, respectively, i.p.) and then placed in a stereotaxic apparatus (David Kopf model 940). A pulled glass micropipette (20–40 μm diameter tip) was used for stereotaxic injections of adeno-associated virus (AAV). Virus was injected into the SON (200 nl/side; AP: −0.65 mm; ML: ±1.25 mm; DV: −5.4 mm from bregma) by an air pressure system using picoliter air puffs through a solenoid valve (Clippard EV 24VDC) pulsed by a Grass S48 stimulator to control injection speed (40 nl/min). The pipette was removed 3 min post-injection followed by wound closure using tissue adhesive (3 M Vetbond). For viral injections into the VLM, mice were placed into a stereotaxic apparatus with the head angled down at approximately 45°. An incision was made at the level of the cisterna magna, then skin and muscle were retracted to expose the dura mater covering the fourth ventricle. A 28-gauge needle was used to make an incision in the dura and allow access to the VLM. Virus was then injected into the VLM (50 nl*2/side; AP: −0.3 and −0.6 mm; ML: ±1.3 mm; DV: −1.7 mm from obex) as described above. The pipette was removed 3 min post-injection followed by wound closure using absorbable suture for muscle and silk suture for skin. For fiber photometry, an optic fiber (200 μm diameter, NA=0.39, metal ferrule, Thorlabs) was implanted in the SON and secured to the skull with dental cement. Subcutaneous injection of sustained release Meloxicam (4 mg/kg) was provided as postoperative care. The mouse was kept in a warm environment and closely monitored until resuming normal activity. Chemogenetic experiments utilized AAV8-hSyn-DIO-hM3Dq-mCherry (Addgene: 44361) and AAV8-nEF-fDIO-hM4Di-mCherry (custom-made vector) produced from Boston Children's Hospital Viral Core and AAV5-EF1α-DIO-mCherry purchased from the UNC Vector Core. Fiber photometry experiments were conducted using AAV1-hSyn-FLEX-GCaMP6s purchased from the University of Pennsylvania (School of Medicine Vector Core). Projection mapping and ChR2-assisted circuit mapping were done using AAV9-EF1α-DIO-ChR2(H134R)-mCherry purchased from the University of Pennsylvania (School of Medicine Vector Core).

## Fiber photometry experiments and analysis of photometry data

In vivo fiber photometry was conducted as previously described (*Mandelblat-Cerf et al., 2017*). A fiber optic cable (1 m long, metal ferrule, 400 μm diameter; Doric Lenses) was attached to the implanted optic cannula with zirconia sleeves (Doric Lenses). Laser light (473 nm) was focused on the opposite end of the fiber optic cable to titrate the light intensity entering the brain to 0.1–0.2 mW.

Emitted light was passed through a dichroic mirror (Di02-R488−25 × 36, Semrock) and GFP emission filter (FF03-525/50-25, Semrock), before being focused onto a sensitive photodetector (Newport part #2151). The GCaMP6 signal was passed through a low-pass filter (50 Hz), and digitized at 1 kHz using a National Instruments data acquisition card and MATLAB software.

All experiments were conducted in the home-cage in freely moving mice. Animals prepared for in vivo fiber photometry experiments (outlined above), were subjected to an ITT or 2DG injection after overnight fasting. Prior to insulin or 2DG injection, a period of GCaMP6s activity was recorded (3 min) to establish baseline activity. Insulin (i.p. 2 U/kg), 2DG (i.p. 500 mg/kg), or saline vehicle was then administered, and GCaMP6 activity was recorded for a further 40 min. In some experiments, mice were pre-treated i.p. with CNO (1 mg/kg) or saline, 30 min prior to insulin or 2DG. The recorded data were exported and then imported into MATLAB for analysis. Fluorescent traces were down-sampled to 1 Hz and the signal was normalized to the baseline ($F_0$ mean activity during baseline activity), with 100% signal being defined as the maximum signal in the entire trace (excluding the injection artifact). Following the ITT, the signal was binned (1 min) and a mean for each bin was calculated. These binned signals were compared to baseline signal using a one-way RM ANOVA.

## Surgery for continuous glucose monitoring

Animals that have undergone fiber photometry surgeries (3 weeks prior) were anesthetized and maintained with isoflurane. Once mice were fully anesthetized, the ventral abdomen and underside of the neck were shaved and disinfected. Animals were placed on their backs on a heated surgical surface. For transmitter implantation, a ventral midline abdomen incision was made and the abdominal wall was incised. The transmitter was placed in the abdominal cavity with the lead exiting cranially and the sensor and connector board exteriorized. The incision was sutured incorporating the suture rib into the closure. For glucose probe implantation, a midline neck incision was performed and the left common carotid artery was isolated. The vessel was then perforated and the sensor of the glucose probe (HD-XG, Data Sciences International) was advanced into the artery towards the heart, within a final placement in the aortic arch. Once in place, the catheter was secured by tying the suture around the catheter and vessel, and overlying opening in tissue was closed. Mice were kept warm on a heating pad and monitored closely until fully recovered from anesthesia.

## Simultaneous AVP fiber photometry and continuous glucose monitoring

All experiments were conducted in the home-cage in freely moving mice. Animals prepared for in vivo fiber photometry and CGM (outlined above), were subjected to an ITT after overnight fasting. After establishing >3 min of baseline activity, insulin (i.p. 1 U/kg or 1.5 U/kg) or saline vehicle was administered. GCaMP6s activity and blood glucose were recorded throughout 2 hr of experiment. Each recording was separated by at least 48 hr. GCaMP6s recording was performed as described above. Blood glucose was acquired using Dataquest A.R.T. 4.36 system and analyzed using MATLAB. Calibration of HD-XG device was performed as per the manufacturer's manual.

## In vivo measurements of plasma glucose, glucagon, and copeptin

Samples for blood glucose and plasma glucagon measurements were taken from mice in response to different metabolic challenges (described in detail below). Both sexes were used for these experiments. Blood glucose was measured with an Accu-Chek Aviva (Roche Diagnostic, UK) and OneTouch Ultra (LifeScan, UK). Plasma copeptin in mouse was measured using an ELISA (MyBioSource, USA and Neo Scientific, USA). We note that the Kryptor BRAHMs system used for human samples could not be used for mouse samples (minimum sample volume of 250 µl plasma).

## Insulin tolerance test

Mice were restrained and a tail vein sample of blood was used to measure fed plasma glucose. A further sample was extracted into EDTA coated tubes for glucagon measurements. Aprotinin (1:5, 4 TIU/ml; Sigma-Aldrich, UK) was added to all blood samples. These blood samples were kept on ice until the end of the experiment. Mice were first administered with any necessary pretreatment and then individually caged. Pretreatments included SSR149415 (30 mg/kg in phosphate-buffered saline [PBS] with 5% DMSO and 5% Cremophor EL), LY2409021 (5 mg/kg in PBS with 5% DMSO), CNO (1–3 mg/kg in PBS with 5% DMSO), or the appropriate vehicle. After a 30-min period, mice

were restrained again, and blood was taken via a tail vein or submandibular bleed. This was used for blood glucose measurements, and also for glucagon. Insulin (0.75, 1, or 1.5 U/kg) was then administered i.p., and the mice were re-caged. At regular time intervals after the insulin injection, mice were restrained and a blood sample was extracted. Blood glucose was measured, and blood was taken for glucagon measurements. At the end of the experiment, blood samples were centrifuged at 2700 rpm for 10 min at 4°C to obtain plasma. The plasma was then removed and stored at –80°C. Plasma glucagon measurements were conducted using the 10 µl glucagon assay system (Mercodia, Upsala, Sweden), according to the manufacturer's protocol.

### Glucoprivic response to 2-Deoxy-D-glucose

Wild-type mice were used for 2-Deoxy-D-glucose (2DG) experiments. The mice were single housed 1 week prior to experimental manipulation. On the experimental day, food was removed 4 hr prior to the experiment. 2DG (500 mg/kg) or saline vehicle was then administered i.p., and blood samples were taken at regular intervals for blood glucose and plasma glucagon measurements. In some cohorts, the V1bR antagonist SSR149415 (30 mg/kg in PBS with 5% DMSO and 5% Tween 80), glucagon receptor antagonist LY240901 (5 mg/kg in PBS with 5% DMSO and 5% Tween 80) or appropriate vehicle was administered i.p. 30 min prior to administration of 2DG. Plasma glucagon was measured as described above.

### Brain slice electrophysiology

To prepare brain slices for electrophysiological recordings, brains were removed from anesthetized mice (4–8 weeks old) and immediately placed in ice-cold cutting solution consisting of (in mM): 72 sucrose, 83 NaCl, 2.5 KCl, 1 $NaH_2PO_4$, 26 $NaHCO_3$, 22 glucose, 5 $MgCl_2$, 1 $CaCl_2$, oxygenated with 95% $O_2$ /5% $CO_2$, and measured osmolarity 310–320 mOsm/l. Cutting solution was prepared and used within 72 hr. 250-µm-thick coronal sections containing the PVH and SON were cut with a vibratome (7000smz2-Campden Instruments) and incubated in an oxygenated cutting solution at 34°C for 25 min. Slices were transferred to oxygenated aCSF (126 mM NaCl, 21.4 mM $NaHCO_3$, 2.5 mM KCl, 1.2 mM $NaH_2PO_4$, 1.2 mM $MgCl_2$, 2.4 mM $CaCl_2$, and 10 mM glucose) and stored in the same solution at room temperature (20–24°C) for at least 60 min prior to recording. A single slice was placed in the recording chamber where it was continuously super-fused at a rate of 3–4 ml per min with oxygenated aCSF. Neurons were visualized with an upright microscope equipped with infrared-differential interference contrast and fluorescence optics. Borosilicate glass microelectrodes (5–7 MΩ) were filled with internal solution. All recordings were made using Multiclamp 700B amplifier, and data were filtered at 2 kHz and digitized at 10 kHz. All analysis was conducted off-line in MATLAB.

### Channelrhodopsin-2 assisted circuit mapping (CRACM) of connections from A1/C1 neurons to the SON

A Cre-dependent viral vector containing the light-gated ion channel channelrhodopsin-2 (AAV-DIO-ChR2-mCherry) was injected into the VLM (targeting A1/C1 neurons) of $Avp^{GFP/+}::Th^{Cre/+}$ mice (see *Figure 5a*) as described above (see 'Stereotaxic surgery and viral injections'). Brain slices were prepared (as above) from these mice. The SON was located by using the bifurcation of the anterior and middle cerebral arteries on the ventral surface of the brain as a landmark. Sections of 250 µm thickness of the SON were then cut with a Leica VT1000S or Campden Instrument 7000smz-2 vibratome, and incubated in oxygenated aCSF (126 mM NaCl, 21.4 mM $NaHCO_3$, 2.5 mM KCl, 1.2 mM $NaH_2PO_4$, 1.2 mM $MgCl_2$, 2.4 mM $CaCl_2$, and 10 mM glucose) at 34°C for 25 min. Slices were recovered for 1 hr at room temperature (20–24°C) prior to recording. Whole-cell voltage-clamp recordings were obtained using a Cs-based internal solution containing (in mM): 135 $CsMeSO_3$, 10 HEPES, 1 EGTA, 4 $MgCl_2$, 4 $Na_2$-ATP, 0.4 $Na_2$-GTP, 10 $Na_2$-phosphocreatine (pH 7.3; 295 mOsm). To photostimulate ChR2-positive A1/C1 fibers, an LED light source (473 nm) was used. The blue light was focused on to the back aperture of the microscope objective, producing a wide-field exposure around the recorded cell of 1 mW. The light power at the specimen was measured using an optical power meter PM100D (Thorlabs). The light output is controlled by a programmable pulse stimulator, Master-8 (AMPI Co. Israel) and the pClamp 10.2 software (AXON Instruments).

## Activation of hM3Dq with CNO in AVP neurons

The modified human M3 muscarinic receptor hM3Dq (*Alexander et al., 2009*) was expressed in AVP neurons by injecting a Cre-dependent virus-containing hM3Dq (AAV-DIO-hM3Dq-mCherry) into the SON of mice bearing *Avp* promoter-driven Cre recombinase (*Avp$^{ires-Cre/+}$* mice; see *Figure 1a*). The intracellular solution for current-clamp recordings contained the following (in mM): 128 K gluconate, 10 KCl, 10 HEPES, 1 EGTA, 1 MgCl$_2$, 0.3 CaCl2, 5 Na2ATP, 0.3 NaGTP, adjusted to pH 7.3 with KOH.

## Brain immunohistochemistry

Mice were terminally anesthetized with chloral hydrate (Sigma-Aldrich) and trans-cardially perfused with PBS followed by 10% neutral buffered formalin (Thermo Fisher Scientific). Brains were extracted, cryoprotected in 20% sucrose, sectioned coronally on a freezing sliding microtome (Leica Biosystems) at 40 µm thickness, and collected in two equal series. Brain sections were washed in 0.1 M PBS with Tween-20, pH 7.4 (PBST), blocked in 3% normal donkey serum/0.25% Triton X-100 in PBS for 1 hr at room temperature and then incubated overnight at room temperature in blocking solution containing primary antiserum (rat anti-mCherry, Invitrogen M11217, 1:1,000; chicken anti-GFP, Life Technologies A10262, 1:1000; rabbit anti-vasopressin, Sigma-Aldrich AB1565, 1:1000; rabbit anti-TH, Millipore AB152, 1:1000). The next morning, sections were extensively washed in PBS and then incubated in Alexa-fluor secondary antibody (1:1000) for 2 hr at room temperature. After several washes in PBS, sections were incubated in DAPI solution (1 µg/ml in PBS) for 30 min. Then, sections were mounted on gelatin-coated slides and fluorescence images were captured using an Olympus VS120 slide scanner.

## Reagents

AVP, hydrocortisone and adrenaline were all from Sigma (Sigma-Aldrich, UK). The G$_q$ inhibitor YM-254890 (*Takasaki et al., 2004*) was from Wako-Chem (Wake Pure Chemical Corp). The selective V1bR antagonist SSR149415 (nelivaptan; *Serradeil-Le Gal et al., 2002*) was from Tocris (Bio-Techne Ltd, UK). The glucagon receptor antagonist LY2409021 (adomeglivant; *Kazda et al., 2016*) was from MedKoo Biosciences (USA).

## Clamping studies in human participants

Clamping studies were conducted at Gentofte Hospital, University of Copenhagen. The studies were approved by the Scientific-Ethical Committee of the Capital Region of Denmark (registration no. H-D-2009-0078) and was conducted according to the principles of the Declaration of Helsinki (fifth revision, Edinburgh, 2000).

## Comparison of copeptin in healthy subjects undergoing a hypoglycemic and euglycemic clamp

Samples from the 'saline arm' from 10 male subjects enrolled in an ongoing, unpublished clinical trial (https://clinicaltrials.gov/ct2/show/NCT03954873) were used to compare copeptin secretion during euglycemia and hypoglycemia.

For the study, two cannulae were inserted bilaterally into the cubital veins for infusions and blood sampling, respectively. For the euglycemic study, participants were monitored during fasting glucose levels. For the hypoglycemic clamp, an intravenous insulin (Actrapid; Novo Nordisk, Bagsværd, Denmark, 1.5 mU·kg$^{-1}$ min$^{-1}$) infusion was initiated at time 0 min to lower plasma glucose. Plasma glucose was measured bedside every 5 min and kept >2.2 mM. Arterialised venous blood was drawn at regular time intervals prior to and during insulin infusion.

## Comparison of copeptin in subjects with T1DM and healthy controls

Samples from 20 male (n=10 control and n=10 T1DM patients) from the 'saline arms' of two previously published studies (*Christensen et al., 2011*; *Christensen et al., 2015*) were used to compare copeptin secretion during a hypoglycemic clamp between T1DM and control subjects. The samples from healthy individuals (Controls) were from *Christensen et al., 2011*. These 10 healthy male subjects were of age 23±1 years, BMI 23±0.5 kg/m$^2$, and HbA$_{1c}$ 5.5±0.1%. The T1DM patient samples were from *Christensen et al., 2015*. These patients were; C-peptide negative, age 26±1 years, BMI 24±0.5 kg/m$^2$, HbA$_{1c}$ 7.3±0.2%, positive islet cell, and/or GAD-65 antibodies, treated with multiple doses of insulin (N=9) or insulin pump (N=1), without late diabetes complications, without hypoglycemia

unawareness, and without residual β-cell function (i.e., C-peptide negative after a 5 g arginine stimulation test). For the study, a hypoglycemic clamp was conducted as outlined above. In control subjects, insulin was infused at a rate of 1.5 mU·kg⁻¹ min⁻¹ for the entire study period (90 min). In subjects with T1D, insulin was infused at a rate of 1 mU·kg⁻¹ min⁻¹ for 60 min (in addition to their regular basal insulin treatment).

### Measurement of copeptin, glucagon, and AVP in human plasma

Copeptin in human plasma was analyzed using the KRYPTOR compact PLUS (Brahms Instruments, Thermo Fisher Scientific, DE). Glucagon was measured using human glucagon ELISA from Mercodia. Plasma AVP was measured using a human AVP ELISA Kit (CSB-E09080h; Cusabio, China).

### Statistical tests of data

All data are reported as mean ± SEM. Unless otherwise stated, N refers to the number of mice. Statistical significance was defined as $p < 0.05$. All statistical tests were conducted in Prism8 (GraphPad Software, San Diego, CA). For two groupings, a t-test was conducted with the appropriate post hoc test. For more than two groupings, a one-way ANOVA was conducted (repeated measures [RM], if appropriate). If data were separated by two treatments/factors, then a two-way ANOVA was conducted. An RM two-way ANOVA was used (if appropriate), and a mixed-models ANOVA was used in the event of an RM with missing data.

## Acknowledgements

The authors would like to thank Prof. W Scott Young and Emily Shepard from NIMH for kindly providing us with *Avpr1b* knockout mice and Professor Guy A Rutter for hosting eye imaging experiments in his laboratory at Imperial College, UK. The authors thank the Alberta Diabetes Institute IsletCore (University of Alberta, AB, Canada) for providing human islets, the isolation of which was supported in part by the Alberta Diabetes Foundation, the Human Organ Procurement and Exchange Program (Edmonton), and the Trillium Gift of Life Network (Toronto). The authors also thank Prof. Paul RV Johnson and colleagues at the Diabetes Research and Wellness Foundation Islet Isolation Facility. The authors would also like to thank the generosity of the organ donors and their families. This study was funded by the following: Wellcome Senior Investigator Award (095531), Wellcome Strategic Award (884655), Sir Henry Wellcome Postdoctoral Fellowship (Wellcome, 201325/Z/16/Z), European Research Council (322620), Leona M and Harry B Helmsley Charitable Trust, Swedish Research Council, Swedish Diabetes Foundation, JRF from Trinity College, Goodger & Schorstein Scholarship (2017), and the Knut and Alice Wallenberg's Stiftelse. JGK and TGH were/are supported by a Novo Nordisk postdoctoral fellowship run in partnership with the University of Oxford. JAK held DPhil studentship from the OXION Programme (Wellcome). PEM holds a grant from the Canadian Institutes of Health Research (CIHR: 148451). BBL is the recipient of grants from the NIH (R01 DK075632, R01 DK089044, R01 DK111401, R01 DK096010, P30 DK046200, and P30 DK057521). AK holds an NIH grant (F31 DK109575). VS is a Diabetes UK Harry Keen Clinical Fellow. IWA holds grants from the Swedish Research Council (2020-01463), the Swedish Diabetes Foundation (DIA2016-127), the Novo Nordisk Foundation (NNF19OC0056601), and the Diabetes Research & Wellness Foundation (2334).

## Additional information

### Funding

| Funder | Grant reference number | Author |
| --- | --- | --- |
| Wellcome Trust | 201325/Z/16/Z | Linford JB Briant |
| H2020 European Research Council | 322620 | Patrik Rorsman |
| Leona M. and Harry B. Helmsley Charitable Trust | | Patrik Rorsman<br>Linford JB Briant |

| Funder | Grant reference number | Author |
| --- | --- | --- |
| Vetenskapsrådet | | Patrik Rorsman |
| Wellcome Trust | 095531 | Patrik Rorsman |
| Wellcome Trust | 884655 | Patrik Rorsman |
| Canadian Institutes of Health Research | 148451 | Patrick E Macdonald |
| Knut och Alice Wallenbergs Stiftelse | | Patrik Rorsman |
| National Institutes of Health | R01 DK075632 | Bradford B Lowell |
| National Institutes of Health | F31 DK109575 | Angela Kim |
| Diabetes UK | Harry Keen Clinical Fellowship | Victoria Salem |
| Diabetesförbundet | | Patrik Rorsman |
| National Institutes of Health | R01 DK089044 | Bradford B Lowell |
| National Institutes of Health | R01 DK111401 | Bradford B Lowell |
| National Institutes of Health | R01 DK096010 | Bradford B Lowell |
| National Institutes of Health | P30 DK046200 | Bradford B Lowell |
| National Institutes of Health | P30 DK057521 | Bradford B Lowell |
| Swedish Diabetes Foundation | DIA2019-419 | Ingrid Wernstedt Asterholm |
| Swedish Research Council | 2020-01463 | Ingrid Wernstedt Asterholm |
| Novo Nordisk Foundation | NNF19OC0056601 | Ingrid Wernstedt Asterholm |
| Diabetes Research and Wellness Foundation | 2334 | Ingrid Wernstedt Asterholm |
| Helmsley Charitable Trust | Grant #1912-03551 | Filip K Knop |
| Novo Nordisk Foundation | 2008P_A11351 [3023] | Filip K Knop |

The funders had no role in study design, data collection and interpretation, or the decision to submit the work for publication.

## Author contributions

Angela Kim, Conceptualization, Data curation, Formal analysis, Investigation, Methodology, Project administration, Resources, Validation, Visualization, Writing – review and editing; Jakob G Knudsen, Lisa Mellander, Haopeng Lin, Data curation, Formal analysis, Investigation, Methodology, Writing – review and editing; Joseph C Madara, Caroline Miranda, Data curation, Formal analysis, Investigation, Methodology, Visualization, Writing – review and editing; Anna Benrick, Data curation, Formal analysis, Investigation, Visualization, Writing – review and editing; Thomas G Hill, Data curation, Investigation, Methodology, Writing – review and editing; Lina Abdul Kadir, Data curation, Investigation, Writing – review and editing; Joely A Kellard, Data curation, Investigation, Methodology, Validation, Writing – review and editing; Timothy James, Methodology, Resources, Validation, Writing – review and editing; Kinga Suba, Yanling Wu, Data curation, Investigation, Methodology, Visualization, Writing – review and editing; Aliya F Spigelman, Data curation, Investigation, Methodology, Validation, Visualization, Writing – review and editing; Patrick E MacDonald, Data curation, Formal analysis, Funding acquisition, Investigation, Methodology, Project administration, Resources, Writing – review and editing;

Ingrid Wernstedt Asterholm, Conceptualization, Data curation, Formal analysis, Funding acquisition, Investigation, Methodology, Resources, Supervision, Validation, Writing – review and editing; Tore Magnussen, Data curation, Formal analysis, Investigation, Methodology, Project administration, Validation, Writing – review and editing; Mikkel Christensen, Tina Vilsbøll, Data curation, Formal analysis, Funding acquisition, Investigation, Methodology, Project administration, Resources, Supervision, Validation, Writing – review and editing; Victoria Salem, Conceptualization, Data curation, Formal analysis, Funding acquisition, Investigation, Methodology, Project administration, Resources, Supervision, Validation, Visualization, Writing – review and editing; Filip K Knop, Conceptualization, Data curation, Funding acquisition, Investigation, Methodology, Project administration, Resources, Supervision, Validation, Writing – original draft, Writing – review and editing; Patrik Rorsman, Data curation, Formal analysis, Funding acquisition, Investigation, Methodology, Project administration, Resources, Supervision, Validation, Writing – original draft, Writing – review and editing; Bradford B Lowell, Conceptualization, Data curation, Funding acquisition, Investigation, Methodology, Project administration, Resources, Software, Supervision, Validation, Writing – original draft, Writing – review and editing; Linford JB Briant, Conceptualization, Data curation, Formal analysis, Funding acquisition, Investigation, Methodology, Project administration, Resources, Supervision, Validation, Visualization, Writing – original draft, Writing – review and editing

### Author ORCIDs
Angela Kim  http://orcid.org/0000-0002-9475-0798
Anna Benrick  http://orcid.org/0000-0003-4616-6789
Lina Abdul Kadir  http://orcid.org/0000-0001-6440-1282
Joely A Kellard  http://orcid.org/0000-0002-0822-1460
Ingrid Wernstedt Asterholm  http://orcid.org/0000-0002-0755-5784
Patrik Rorsman  http://orcid.org/0000-0001-7578-0767
Bradford B Lowell  http://orcid.org/0000-0002-0436-3760
Linford JB Briant  http://orcid.org/0000-0003-3619-3177

### Ethics

Human subjects: Ethics statements are given in the Methods as follows: Clamping studies were conducted at Gentofte Hospital, University of Copenhagen. The studies were approved by the Scientific-Ethical Committee of the Capital Region of Denmark (registration no. H-D-2009-0078 and H-18046965) and was conducted according to the principles of the Declaration of Helsinki (fifth revision, Edinburgh, 2000). Human pancreatic islets were isolated, with ethical approval and clinical consent, at the Diabetes Research and Wellness Foundation Human Islet Isolation Facility (OCDEM, Oxford, UK) or Alberta Diabetes Institute IsletCore (University of Alberta, AB, Canada).

All animal experiments were conducted in strict accordance to regulations enforced by the research institution. Experiments conducted in the UK were done so in accordance with the UK Animals Scientific Procedures Act (1986) (P0A6474AE, P0FA927F8) and University of Oxford and Imperial College London ethical guidelines, and were approved by the local Ethical Committee. All animal care and experimental procedures conducted in the U.S.A. were approved by the Beth Israel Deaconess Medical Center Institutional Animal Care and Use Committee. Animal experiments conducted in Goteborg University were approved by a local Ethics Committee.

### Decision letter and Author response
Decision letter https://doi.org/10.7554/eLife.72919.sa1
 https://doi.org/10.7554/eLife.72919.sa2

## Additional files

### Supplementary files
• Transparent reporting form
• Source data 1. Data for figures.

### Data availability
The study data are available as a supplementary file.

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
