## [Editor Report]

The authors have revised their manuscript in response to the comments and suggestions by the two reviewers. The current study provides compelling data to support a mechanistic model that the CNS regulates glucagon secretion through glucose-regulated AVP release.

---

## [Decision Letter]

**Decision letter after peer review:**

Thank you for submitting your article "Arginine-vasopressin mediates counter-regulatory glucagon release and is diminished in type 1 diabetes" for consideration by *eLife*. Your article has been reviewed by 2 peer reviewers, and the evaluation has been overseen by a Reviewing Editor and Lu Chen as the Senior Editor. The reviewers have opted to remain anonymous.

Essential revisions:

The comments of both reviewers can be addressed via revisions/additions to the text and figures without new experiments.

*Reviewer #1 (Recommendations for the authors):*

---

## [Author Response]

In this manuscript, Angela Kim et al., use a combination of in vitro and in vivo studies to determine how glucose-control of central AVP release controls pancreatic α-cell calcium influx and glucagon secretion to modulate blood glucose homeostasis. The manuscript clearly shows that activation of AVP release from magnocellular AVP neurons stimulates pancreatic islet glucagon secretion. Furthermore, the manuscript finds AVP (measured by circulating Copeptin) is elevated in plasma following insulin induced hypoglycemia, which also activates AVP neuron electrical excitability and calcium entry. To confirm that AVP release stimulates glucagon secretion via islet α-cell Avpr1b activation, both Avpr1b antagonists and an Avpr1b-/- mouse model were utilized. Finally, the manuscript looks at plasma AVP in humans undergoing a hypoglycemic clamp; while this results in AVP release in non-diabetic controls, AVP release is blunted following hypoglycemia in type-1 diabetic patients. Based on an extensive amount of high-quality data, the authors conclude that AVP release from magnocellular AVP neurons is involved in regulating glucagon secretion in response to hypoglycemia. The manuscript is well written and easy to follow. As the exact mechanism that controls glucagon secretion is still unknown, this manuscript adds important information for the diabetes research community detailing the importance of CNS control of islet glucagon secretion through glucose regulated AVP release. Overall, this is an excellent manuscript that will be very useful to the diabetes research community.